# On the Adversarial Robustness of Mixture of Experts

**Joan Puigcerver**
Google Research

**Rodolphe Jenatton**
Google Research

**Carlos Riquelme**
Google Research

**Pranjal Awasthi**
Google Research

**Srinadh Bhojanapalli**
Google Research

## Abstract

Adversarial robustness is a key desirable property of neural networks. It has been empirically shown to be affected by their sizes, with larger networks being typically more robust. Recently, Bubeck and Sellke [3] proved a lower bound on the Lipschitz constant of functions that fit the training data in terms of their number of parameters. This raises an interesting open question, do—and can—functions with more parameters, but not necessarily more computational cost, have better robustness? We study this question for sparse Mixture of Expert models (MoEs), that make it possible to scale up the model size for a roughly constant computational cost. We theoretically show that under certain conditions on the routing and the structure of the data, MoEs can have significantly smaller Lipschitz constants than their dense counterparts. The robustness of MoEs can suffer when the highest weighted experts for an input implement sufficiently different functions. We next empirically evaluate the robustness of MoEs on ImageNet using adversarial attacks and show they are indeed more robust than dense models with the same computational cost. We make key observations showing the robustness of MoEs to the choice of experts, highlighting the redundancy of experts in models trained in practice.

## 1 Introduction

Adversarial robustness refers to prediction robustness of a given machine learning model to adversarial, but bounded, changes to the input. Neural networks trained with standard classification objectives have been shown to have poor adversarial robustness [25], a property attributed to their overparametrization. Conversely, in practice, larger models, that have more parameters and higher computation cost, have shown better robustness [20, 26, 1, 12].

In a recent work, Bubeck and Sellke [3] studied this phenomenon from a theoretical perspective, by analyzing the relationship between model size and its Lipschitz constant - which measures the sensitivity of a function to changes in the input. In particular the authors proved that any function with $P$ parameters, that memorizes $N$ input data points in $D$ dimensions, has a Lipschitz constant of at least $\mathcal{O}(\sqrt{\frac{N \cdot D}{P}})$. This shows that, on a given dataset, larger models can have better robustness (smaller Lipschitz constant). Note that as this is only a lower bound, it does not guarantee that larger models will indeed have a smaller Lipschitz constant. Interestingly, the result is agnostic to other properties of the function, such as its computation cost, or specific architecture.

Given the above result it is natural to wonder: can one increase the model size, without increasing the computation cost, and enjoy better robustness? An increasingly popular class of such models are Mixture of Experts (MoE) [23]. MoE models have multiple expert sub-models, and a routing function that selects for each input a small subset of the experts and routes the input only to them. Neural networks with MoE layers, such as Switch Transformer [10] in NLP, and V-MoE [22] in Computer Vision, have been shown to achieve superior performance in comparison to their dense counterparts,

36th Conference on Neural Information Processing Systems (NeurIPS 2022).

by allowing one to scale the model size without increasing the computation cost. In this paper we study the following question: are MoE models more adversarially robust than dense models?

In general, sparse MoE models are not continuous, and hence not even smooth. Small changes to the input can result in the router selecting a different expert for a given example. Unfortunately, in some cases these experts may be very different, resulting in large changes to the output. Another factor that affects the robustness of MoE models is the geometry of input data and its routing –how the data is divided between experts. This decides what data an expert is trained on and hence their robustness. Finally, it has been observed that –unless certain auxiliary losses are also applied to encourage balancedness– the number of used experts tends to collapse to very few, and the remaining ones are just ignored [22, 10]. Given these stability issues, it is a priori *not* clear if MoEs, despite having more parameters, are more robust than dense models.

To theoretically study the effect of both of these factors, (1) router stability and (2) data routing, on robustness we consider the *smooth* MoE models, where each expert output is weighted by their routing probability. In particular, MoE models where all experts are simultaneously applied to every input. We analyze models with fixed routing proving that MoEs with *linear experts* can achieve better robustness if the data is well separated and routed accordingly. In the extreme case, MoEs with $E$ experts can have a smaller Lipschitz constant by a factor of $1/\sqrt{E}$ compared to equivalent dense models. We also characterize the effect of the difference between experts in terms of robustness, showing that MoEs can have a high Lipschitz constant when two experts are very different for inputs that weigh the experts similarly and highly. We show that both these factors, data routing and difference between relevant experts, characterize the Lipschitz constant of MoE models.

We next evaluate robustness experimentally by using adversarial attacks [20]. We experimentally show that on the ImageNet dataset [6], MoEs are more robust than dense models that have the same computation cost. Interestingly, we observe that adversarial attacks result in significant routing changes of the inputs, but do not result in lower robustness than dense models. This suggests that in practice, MoEs are robust to the choice of the experts to an extent. We next perform standard adversarial training of both the dense and the MoE models and again observe that MoEs are more robust against adversarial attacks.

The main contributions of the current work are as follows:

1. We propose a simple theoretical framework to understand the robustness of Mixture of Experts (MoE) models. We provide general and non-trivial sufficient conditions under which MoEs are provably more robust than their dense counterparts for the linear experts setting.

2. We perform extensive experiments to demonstrate that in practice MoEs indeed enjoy better robustness than dense models to norm bounded adversarial attacks. We also uncover intriguing properties of adversarial attacks for MoE models and show that even for robust MoE models, the attacks very often change the routing of data points, thereby pointing to a high degree of redundancy in such models.

## 2 Preliminaries

### 2.1 Mixture of Experts

Mixture of Experts combine the outputs of (sub-)models, i.e., the *experts*, via a weighted sum of their outputs [14, 15, 28, 9]. *Sparse* Mixture of Experts condition the weights of the sum in the inputs, and activate only $K$ (out of $E$) experts, where $K$ is typically a very small integer compared to $E$ (such as $K = 1$ or $K = 2$) [23, 10]. This form of conditional computation models allows one to easily increase the number of parameters in the model (roughly) independently of its compute cost [21]. This approach has been recently applied to significantly increase the model size and quality of models used in Natural Language Processing [23, 18, 19, 10, 8] and Computer Vision applications [22, 27].

More concretely, we define the sparse and smooth (or dense) versions of mixtures of experts below. While previous empirical works have sometimes selected more than one expert per input ($K > 1$), for simplicity we constrain ourselves to the case where only one expert is selected for each input ($K = 1$). Recently, practical models also tend to this setup to match the FLOPs of dense models [10].

**Sparse MoEs.** In these models only one expert is selected for a given example $x$ based on its routing probabilities $p_i(x)$. These models are not continuous functions of the input. A small change in the input can lead to large changes in the output, due to changes in selected experts. A sparse MoE layer

is defined as:

$$f(\boldsymbol{x}) = \sum_{i=1}^{E} \mathbf{1}_{\{p_i(\boldsymbol{x}) \geq p_j(\boldsymbol{x}), \forall j \neq i\}} \ f_i(\boldsymbol{x}). \tag{1}$$

Here $f_i(\boldsymbol{x})$ are the individual expert functions. $\mathbf{1}$ is the indicator function that takes the value of 1 if the condition is satisfied and 0 else. Note that to do tie breaking in the above definition, in case multiple experts have the same maximum probability, we simply sample uniformly one of the experts with the maximum probability. In practice, MoE models can have multiple sparse layers in combination with dense layers. Further, random noise is sometimes added to the $p_i$'s [23, 22].

We refer to MoEs with only 1 expert as **dense** models. In dense models, which are the standard in neural networks literature, same function is applied to all input examples. In sparse MoEs introduced above, different part of network / expert is applied to each input example depending on which expert selected. This flexibility allows us to train larger sparse MoEs, that have the same computational cost as the dense models, as only a part of MoE is activated/selected for each input, with better accuracy than dense models [10, 22].

**Smooth MoEs.** To theoretically analyze robustness, we consider a smooth mixture of expert models, as they are continuous functions. In this case, the output of each expert $f_i$, is weighted according to its routing probabilities:

$$f(\boldsymbol{x}) = \sum_{i=1}^{E} p_i(\boldsymbol{x}) f_i(\boldsymbol{x}). \tag{2}$$

The routing probabilities $p_i(\boldsymbol{x})$ are usually computed by a routing layer, e.g., $p_i(\boldsymbol{x}) = \sigma(\boldsymbol{S}\boldsymbol{x})_i$, where $\sigma$ is the softmax and $\boldsymbol{S} \in \mathbb{R}^{E \times D}$ is a trainable variable. Such linear routing layers are the common choice for MoE models in practice [10].

Though these models have recently been shown to achieve state of art performance for tasks in NLP [10] and vision [22], there have not been many works analyzing their robustness. Recently, Allingham et al. [2] empirically studied robustness of MoEs to natural perturbations in data (e.g., evaluation across different corrupted versions of CIFAR10 [17] and ImageNet [6]), and showed MoEs are more robust for these natural perturbations than the corresponding dense models. To the best of our knowledge ours is the first work theoretically analyzing robustness of MoEs, and empirically exploring it in the context of adversarial perturbations.

**Load balancing loss** To prevent MoEs from collapsing and always selecting the same expert for all inputs, it is customary during training to add a *load balancing loss*, that encourages equal fraction of inputs being routed to different experts. We define and present these losses in detail in appendix E.

## 2.2 Adversarial robustness

Adversarial robustness models the susceptibility of a function to adversarial perturbations to its input [25, 20]. More concretely, given a function $f$, loss function $\ell$, and an input $\boldsymbol{x} \in \mathbb{R}^D$, we can write the *adversarial loss* incurred at $\boldsymbol{x}$ with label $\boldsymbol{y}$ as follows.

$$\max_{\boldsymbol{z}:\|\boldsymbol{z}\| \leq \epsilon} \ell(f(\boldsymbol{x} + \boldsymbol{z}), \boldsymbol{y}). \tag{3}$$

Here $\epsilon$ is the attack radius per input. For the norm constraint ($\|\boldsymbol{z}\|$), popular choices in practice, are $\ell_\infty$ and $\ell_2$ norms [20, 4]. Adversarial accuracy is the accuracy of the model on data with perturbations that satisfy the above equation, i.e., when the loss function $\ell$ is the $0/1$ classification loss.

Since optimizing equation 3 for neural networks is computationally hard, popular methods to find adversarial examples use local search approaches such as gradient ascent. The Fast Gradient Signed Method (FGSM) is perhaps the simplest one, with only one gradient update step [11]. For $\ell_\infty$ bounded perturbations the FGSM update is the following:

$$\boldsymbol{x} + \boldsymbol{z} = \boldsymbol{x} + \epsilon \ \mathrm{sgn}\left(\nabla_{\boldsymbol{x}} \ell(f(\boldsymbol{x}), \boldsymbol{y})\right). \tag{4}$$

sgn() is the sign function. The Projected Gradient Descent (PGD) method is a stronger attack that does multiple ($\tau$) gradient ascent steps to find the perturbation [20]. The update rule for $\ell_\infty$ bounded perturbations is as follows, starting with $\boldsymbol{x}^0 = \boldsymbol{x}$.

$$\boldsymbol{x}^{t+1} = \Pi_{\boldsymbol{x}+\mathcal{C}}\left[\boldsymbol{x}^t + \alpha \ \mathrm{sgn}\left(\nabla_{\boldsymbol{x}} \ell(f(\boldsymbol{x}), \boldsymbol{y})\right)\right], \forall t \in [0, \tau-1]. \tag{5}$$

$\alpha$ is chosen to be $\frac{\epsilon}{\tau}$. Here $\mathcal{C}$ is the constraint set $\{\boldsymbol{z} : \|\boldsymbol{z}\|_\infty \le \epsilon\}$ and $\Pi$ is the projection operator.

Existing works have established that neural networks are sensitive to adversarial attacks resulting in a significant drop in their accuracy [25]. Interestingly, increasing the size of neural networks, thereby increasing their capacity, leads to an improvement in adversarial accuracy, showing that larger neural networks are more robust [20, 26, 1, 12]. In a recent work, Bubeck and Sellke [3] studied this phenomenon theoretically. They proved a lower bound on the Lipschitz constant of any function that fits the training data that scales inversely $\frac{1}{\sqrt{P}}$ with the number of function parameters $P$. However it is not clear if functions that have more parameters, but not necessarily more computation cost (e.g. MoEs), can achieve better robustness. In this paper we analyze adversarial robustness of MoEs, both theoretically and experimentally.

**Notation.** We use small bold letters, e.g., $\boldsymbol{x}$, to denote vectors and capital bold letters to denote matrices, e.g., $\boldsymbol{W}$. Scalars are denoted with plain letters. $[N]$ denotes the integer set from 1 to $N$. $\|.\|$ denotes the $\ell_2$ norm unless specified otherwise.

# 3 Robustness analysis

In this section, we present our main results regarding the robustness of the mixture of expert models. We first present a bound on the Lipschitz constant of MoEs for general expert functions $f_i$. While this is an upper bound for general functions it still highlights the two key components that affect the robustness of MoE models.

## 3.1 Router stability

In this section, we compute the Lipschitz constant of smooth MoEs with learnable routing. Let,

$$f(\boldsymbol{x}) = \sum_{i=1}^{E} p_i(\boldsymbol{x}) f_i(\boldsymbol{x}) \ \text{ where } \ p_i(\boldsymbol{x}) = \frac{\exp(\langle \boldsymbol{s}_i, \boldsymbol{x}\rangle)}{\sum_{j=1}^{E} \exp(\langle \boldsymbol{s}_j, \boldsymbol{x}\rangle)} \tag{6}$$

and $\{\boldsymbol{s}_i\}_{i\in[E]}$, with each $\boldsymbol{s}_i$ in $\mathbb{R}^D$, are learnable variables that decide the routing of an example to different experts. We then prove the following upper bound on the Lipschitz constant of the MoE models with learnable routing.

**Lemma 1.** *Let $\{f_i\}_{i\in[E]}$ be smooth functions with Lipschitz constants $\{L_{f_i}\}_{i\in[E]}$ and let $L_f$ be the Lipschitz constant of $f$. Let $\bar{\boldsymbol{s}}(\boldsymbol{x}) = \sum_j p_j(\boldsymbol{x})\boldsymbol{s}_j$. Then $f(\boldsymbol{x})$ in equation 6 satisfies,*

$$\|\nabla_{\boldsymbol{x}} f(\boldsymbol{x})\| \le \sum_{i=1}^{E} p_i(\boldsymbol{x}) L_{f_i} + \left\| \sum_{i=1}^{E} p_i(\boldsymbol{x}) f_i(\boldsymbol{x}) (\boldsymbol{s}_i - \bar{\boldsymbol{s}}(\boldsymbol{x})) \right\|,$$

*and hence*

$$L_f \le \max_{i\in[E]} \{L_{f_i}\} + \sup_{x} \left\| \sum_{i=1}^{E} p_i(\boldsymbol{x}) f_i(\boldsymbol{x}) (\boldsymbol{s}_i - \bar{\boldsymbol{s}}(\boldsymbol{x})) \right\|. \tag{7}$$

The above Lemma bounds the Lipschitz constant of MoE models with two terms that depend on (1) data routing, and (2) router stability. The first term above is from the individual Lipschitz constant of the experts. This depends on how the data is partitioned or routed to different experts, which we will discuss in more detail in the next section. The second term arises from the router used to compute probabilities for different experts. One may wonder if we can make this term arbitrarily large by increasing the norm of $\boldsymbol{x}$. This is not the case as increasing the norm of $\boldsymbol{x}$ usually results in the collapse of the routing probabilities to a single expert. This makes $p_i(\boldsymbol{x})(\boldsymbol{s}_i - \bar{\boldsymbol{s}}(\boldsymbol{x}))$ zero for all experts $i \in [E]$.

To study the above more concretely, we consider the case of $E = 2$ experts. In this setting the second term above reduces to $\|p_1(\boldsymbol{x}) f_1(\boldsymbol{x})(\boldsymbol{s}_1 - \bar{\boldsymbol{s}}(\boldsymbol{x})) + (1 - p_1(\boldsymbol{x})) f_2(\boldsymbol{x})(\boldsymbol{s}_2 - \bar{\boldsymbol{s}}(\boldsymbol{x}))\|$. Now, in the extreme case, where $p_1(\boldsymbol{x})$ is either 0 or 1, the above term collapses to 0 as $p_i(\boldsymbol{x}) f_i(\boldsymbol{x})(\boldsymbol{s}_i - \bar{\boldsymbol{s}}(\boldsymbol{x}))$ is 0 in that case. This is also expected as for examples that are routed to an expert with high probability, the main dominating factor is the individual expert Lipschitz constant, as small perturbations do not cause much change to the probabilities.

A more interesting setting is when $p_1(\boldsymbol{x})$ is $\frac{1}{2}$. In this setting, small perturbations to the input can cause the model to drastically change the weight between the experts, and the above term reduces to $\|\frac{1}{4} \cdot (f_1(\boldsymbol{x}) - f_2(\boldsymbol{x})) \cdot (\boldsymbol{s}_1 - \boldsymbol{s}_2)\|$. Hence MoE models can suffer from large Lipschitz constants if the two experts are very different for points on the boundary, i.e., if $|f_1(\boldsymbol{x}) - f_2(\boldsymbol{x})| \gg 0$. Alternately if $f_1(\boldsymbol{x}) \approx f_2(\boldsymbol{x})$ for points on the boundary (i.e. $p_1(\boldsymbol{x}) \approx \frac{1}{2}$) then this term is small.

We will later see in the experiments that MoEs trained in practice often have good router stability and changes to routing do not result in much degradation of the final accuracy.

## 3.2 Data routing

In this section we will study the effect of data routing on the Lipschitz constant of the individual experts – the first term in RHS of equation 7 (lemma 1). In particular we will try to address how large can $\max_{i \in [E]}\{L_{f_i}\}$ be in comparison to the Lipschitz constant of a dense model trained on the same data. Towards this end, we will theoretically analyze the robustness of MoE models with a pre-determined fixed routing and linear experts, in the light of the structure of the data.

### 3.2.1 Setup

We consider the setting of fixed routing, where we assume the routing of individual examples to experts is pre-determined and fixed. Moreover, we consider *linear models* as experts.

More specifically, we consider $N$ input points of dimension $D$ stacked in the matrix $\boldsymbol{X} \in \mathbb{R}^{N \times D}$, together with the corresponding vector of targets $\boldsymbol{y} \in \mathbb{R}^N$. In the dense case (i.e., no experts), we would like to find a linear model $\boldsymbol{w}^* \in \mathbb{R}^D$ that minimizes

$$\min_{\boldsymbol{w} \in \mathbb{R}^D} \|\boldsymbol{X}\boldsymbol{w} - \boldsymbol{y}\|^2. \tag{8}$$

The least-squares solution [13] for this problem that minimizes the MSE loss is $\boldsymbol{w}^* = (\boldsymbol{X}^\top \boldsymbol{X})^\dagger \boldsymbol{X}^\top \boldsymbol{y}$. Here $\dagger$ denotes the pseudo inverse. On the other hand, for mixture of experts, let the dataset be split into $E$ subsets $\mathcal{S}_1, \cdots, \mathcal{S}_E$, with $\boldsymbol{X}_i, \boldsymbol{y}_i$ denoting the data in set $\mathcal{S}_i$. Now let the data from the set $\mathcal{S}_i$ be routed to the expert $i$. Below we write the objective for each expert:

$$\min_{\boldsymbol{w}_i \in \mathbb{R}^D} \|\boldsymbol{X}_i\boldsymbol{w}_i - \boldsymbol{y}_i\|^2 \text{ for } i \in [E]. \tag{9}$$

Similarly, let $\boldsymbol{w}_i^* = (\boldsymbol{X}_i^\top \boldsymbol{X}_i)^\dagger \boldsymbol{X}_i^\top \boldsymbol{y}_i$ be the optimal solution for expert $i$ that minimizes the MSE.

### 3.2.2 Analysis

Before introducing our analysis relating the Lipschitz constant of the dense and MoE models, we first present two examples to better understand how routing affects Lipschitz constants. In particular we consider a simple setting with $E = 2$ experts. Let $\boldsymbol{X}^\top = [\boldsymbol{X}_1^\top, \boldsymbol{X}_2^\top]$, be the data routed to the two experts. Recall that the Lipschitz constant of the function $f(\boldsymbol{X}) = \boldsymbol{X}\boldsymbol{w}$ is $\|\boldsymbol{w}\|$.

**Case $\boldsymbol{X}_1 \perp \boldsymbol{X}_2$.** In this setting we get

$$\|\boldsymbol{w}^*\| = \|(\boldsymbol{X}^\top \boldsymbol{X})^\dagger \boldsymbol{X}^\top \boldsymbol{y}\| = \|(\boldsymbol{X}_1^\top \boldsymbol{X}_1)^\dagger \boldsymbol{X}_1^\top \boldsymbol{y}_1 + (\boldsymbol{X}_2^\top \boldsymbol{X}_2)^\dagger \boldsymbol{X}_2^\top \boldsymbol{y}_2\| = \|\boldsymbol{w}_1^* + \boldsymbol{w}_2^*\|$$
$$\geq \max(\|\boldsymbol{w}_1^*\|, \|\boldsymbol{w}_2^*\|)$$

The second equality follows from the folowing steps - 1) $\boldsymbol{X}^T \boldsymbol{X} = \boldsymbol{X}_1^T \boldsymbol{X}_1 + \boldsymbol{X}_2^T \boldsymbol{X}_2$, 2) $(\boldsymbol{X}_1^T \boldsymbol{X}_1 + \boldsymbol{X}_2^T \boldsymbol{X}_2)^\dagger = (\boldsymbol{X}_1^T \boldsymbol{X}_1)^\dagger + (\boldsymbol{X}_2^T \boldsymbol{X}_2)^\dagger$, and 3) $(\boldsymbol{X}_1^T \boldsymbol{X}_1)^\dagger \boldsymbol{X}_2^T = 0$, where steps 2 and 3 follow from the orthogonality of $\boldsymbol{X}_1$ and $\boldsymbol{X}_2$ (see [16]). Hence, experts have smaller Lipschitz constant when data routed to different experts lies in orthogonal subspaces.

**Case $\boldsymbol{X}_1 = -\boldsymbol{X}_2$ and $\boldsymbol{y}_1 = \boldsymbol{y}_2$.** In this case, we see that:

$$\|\boldsymbol{w}^*\| = \|(\boldsymbol{X}^\top \boldsymbol{X})^\dagger \boldsymbol{X}^\top \boldsymbol{y}\| = \|(\boldsymbol{X}_1^\top \boldsymbol{X}_1 + \boldsymbol{X}_2^\top \boldsymbol{X}_2)^\dagger \boldsymbol{X}^\top \boldsymbol{y}\|$$
$$= \|(\boldsymbol{X}_1^\top \boldsymbol{X}_1 + \boldsymbol{X}_2^\top \boldsymbol{X}_2)^\dagger (\boldsymbol{X}_1^\top \boldsymbol{y}_1 - \boldsymbol{X}_1^\top \boldsymbol{y}_1)\| = 0 \leq \min(\|\boldsymbol{w}_1^*\|, \|\boldsymbol{w}_2^*\|).$$

Hence experts have worse Lipschitz constant when the data routed to different experts is aligned. These two simple examples illustrate under which conditions experts have an advantage over a single dense model, and when they do not.

We now present our main result. To capture this relation between the data geometry and the routing, we introduce the following quantities. Let $\{\boldsymbol{U}_i\}_{i \in [E]}$ be the projection matrices onto orthogonal subspaces in $\mathbb{R}^D$. One way to construct them is by first taking the singular vectors of $\boldsymbol{X}^\top \boldsymbol{X}$ and assigning them to the set $i$ with the largest projection. This guarantees that $\boldsymbol{U}_i$ is orthogonal to $\boldsymbol{U}_j$ with $j \neq i$, and they have greatest alignment with subspace spanned by $\boldsymbol{X}_i$.

We first define a quantity to capture how well $\boldsymbol{U}_i$ captures the span of the data subset $\boldsymbol{X}_i$.

**Definition 1** (In-subspace distance: $\epsilon_1$). $\exists \epsilon_1 \geq 0$, $\|\boldsymbol{U}_i \boldsymbol{U}_i^\top (\boldsymbol{X}^\top \boldsymbol{X})^\dagger - (\boldsymbol{X}_i^\top \boldsymbol{X}_i)^\dagger\|_2 \leq \epsilon_1, \forall i \in [E]$.

Here $\|.\|_2$ for a matrix denotes the spectral norm. $\epsilon_1$ is small when, $\forall i, \boldsymbol{U}_i$ captures $\boldsymbol{X}_i$ perfectly.

Next we define the projection distance between data from two different subsets.

**Definition 2** (Cross-subspace distance: $\epsilon_2$). $\exists \epsilon_2 \geq 0$ *such that for any $\boldsymbol{z}$ in the span of $\boldsymbol{X}_j^\top$, we have* $\|(\boldsymbol{X}_i^\top \boldsymbol{X}_i)^\dagger \boldsymbol{z}\| \leq \epsilon_2 \|\boldsymbol{z}\|, \forall i \neq j$.

Here $\epsilon_2$ is small if the data in different subsets $\boldsymbol{X}_i$ lies in orthogonal subspaces.

**Theorem 1.** *Let $\boldsymbol{w}^*$ be the minimizer of equation 8 and $\{\boldsymbol{w}_i^*\}_{i \in [E]}$ be the minimizers of equation 9. Then,*

$$\|\boldsymbol{w}^*\|^2 \geq \sum_{i=1}^E (\lfloor \|\boldsymbol{w}_i^*\| - \epsilon_1 \|\boldsymbol{X}^\top \boldsymbol{y}\| - \epsilon_2 \sum_{j \neq i} \|\boldsymbol{X}_j^\top \boldsymbol{y}_j\| \rfloor_+)^2,$$

*where $\lfloor \cdot \rfloor_+$ denotes the projection onto non-negative numbers.*

The above result lower bounds the Lipschitz constant of the dense model $\|\boldsymbol{w}^*\|$ in terms of Lipschitz constants of the experts $\|\boldsymbol{w}_i^*\|$ in the MoE model. We present the proof of this theorem in appendix A.

In the case where $\epsilon_1 = \epsilon_2 = 0$, we obtain the following bound $\|\boldsymbol{w}^*\| \geq \sqrt{\sum_{i=1}^E \|\boldsymbol{w}_i^*\|^2}$. Assuming a balanced setting where all experts have the same norm parameters, we get the scaling $\|\boldsymbol{w}^*\| \approx \mathcal{O}(\sqrt{E}) \cdot \max_{i \in [E]}\{\|\boldsymbol{w}_i^*\|\}$. Hence in this setting, experts have a Lipschitz constant that is smaller by a factor of $\frac{1}{\sqrt{E}}$ compared to dense models. This happens when the data lies in $E$ orthogonal subspaces and the data from each subspace is routed to the same expert. This shows that MoE models can have significantly smaller Lipschitz constant than their dense counterparts, while having the same computation cost. For MoEs in practice, data routed to different experts does display some clustering of the features (see Figure 7 in Riquelme et al. [22]).

As data in different partitions $\boldsymbol{X}_i$ gets more aligned, $\epsilon_1$ and $\epsilon_2$ increase, and reduce the gap between the Lipschitz constant of the dense and the MoE models. This reduces the gap in the Lipschitz constant of the dense model and the experts. In the extreme case, if all the datapoints are the same, then $\epsilon_1$ and $\epsilon_2$ are large, eliminating this difference. In appendix D, we give a complementary result relating the Lipschitz constants of the experts to that of the dense model when both $\epsilon_1$ and $\epsilon_2$ can possibly be large.

**Connections to Bubeck and Sellke [3].** Bubeck and Sellke [3] proved a universal lower bound on the Lipschitz constant of a function required to $\delta$-memorize $N$ training samples in $D$ dimension.

$$L_f \geq \tilde{\Omega}\left(\delta \sqrt{\frac{ND}{P}}\right).$$

For the linear model considered above, this reduces to $\tilde{\Omega}\left(\delta\sqrt{N}\right)$ as the number of parameters is $D$. As MoEs have $E$ times more parameters, they have a lower bound of $\tilde{\Omega}\left(\delta\sqrt{\frac{N}{E}}\right)$, i.e., the lower bound on the Lipschitz constant of MoEs is smaller by a factor of $\sqrt{\frac{1}{E}}$. Our result shows that this lower bound is in fact achievable, and hence tight, when the data lies in $E$ orthogonal subspaces and data from each subspace is routed to the same expert.

# 4 Experiments

## 4.1 Setup

We compare the robustness of Vision Transformer (ViT) [7] and Vision MoE (V-MoE) [22] models against adversarial attacks. In particular, we use the ViT-B/32 and V-MoE-B/32 models through all the experiments. These models have the same backbone architecture but the latter replaces one in every two feedforward layers with a *sparse* Mixture of Experts, selecting $K = 2$ out of $E = 32$ feedforward experts that are applied on each token[1]. One could argue that the router in the V-MoE model introduces an overhead that should be accounted for. Thus, we have trained a bigger version of the dense ViT-B/32 (which we coin ViT-B++/32) that reaches roughly the same predictive performance as the V-MoE model, but has higher cost. The cost of evaluating an image on the dense ViT models is 8.9 and 17.9 GFLOPs and runtime cost, respectively; and 12.4 GFLOPs on the V-MoE model. Appendix B contains additional experimental details.

We pre-train our models on the private dataset JFT-300M [24] for 7 epochs ($517\,859$ steps with a batch size of $4\,096$ images), using an image resolution of $224 \times 224$ pixels, and standard data augmentation (inception crop and horizontal flips). Since JFT-300M is a multi-label dataset, we minimize the sigmoid cross-entropy loss. V-MoE also adds auxiliary losses to encourage a balanced load for all experts; we used the same recipe as in [22]. In both cases we use Adam ($\beta_1 = 0.9, \beta_2 = 0.999$), with a peak learning rate of $8 \cdot 10^{-4}$, reached after a linear warm-up of $10^4$ steps and then linearly decayed to a final value of $10^{-5}$. Weight decay of 0.1 was used on all parameters. This is the same pre-training protocol as that used in [7, 22].

After pre-training, the models are fine-tuned on ImageNet [6], at a resolution of $384 \times 384$ pixels and the same data augmentations as before, for a total of $10^4$ steps, using a batch size of $4\,096$ images. SGD with Momentum ($\mu = 0.9$) is used for fine-tuning, with a peak learning rate of $0.03$, reached after a linear warm-up of 500 steps, and followed with cosine decay to a final value of $10^{-5}$. The norm of the flattened vector of gradients is clipped to a maximum value of 10. Since ImageNet images have a single label, we minimize the softmax cross-entropy loss during fine-tuning. The same fine-tuning protocol was adopted in [7, 22].

We evaluate the adversarial robustness of both the pre-trained and fine-tuned models, by means of PGD adversarial attacks [20]. We maximize the corresponding loss (sigmoid or softmax cross-entropy), varying the $\ell_\infty$ norm constraint on the input image, for a total of $\tau = 40$ steps.

## 4.2 Adversarial robustness of V-MoEs

During pre-training the ViT-B/32 model achieves a precision at 1 of 39.3%, and the V-MoE-B/32 achieves a precision-at-1 of 43.5% (conversely, the false discovery rate at 1 is 60.7% and 56.5%, respectively). After fine-tuning, the classification error achieved by each model on ImageNet is 19.3% and 17.8%, respectively.

Figure 1 shows in solid lines the false discovery rate (left) and the classification error rate (right) as a function of the $\ell_\infty$ constraint. Despite the fact that the V-MoE model contains a router that makes discrete choices among the experts conditioned on the input, which could potentially lead to a severe weakness against the adversarial attacker, we can observe that it follows the same trend as the base dense ViT model. It is able to preserve a lower error over a wide range of $\ell_\infty$ values. A larger version of the dense model matching the quality of the V-MoE has a much higher cost.

In addition, we also fine-tuned the base ViT and V-MoE models on ImageNet using PGD adversarial training. We use the same recipe as above but we perform a PGD attack of 10 steps on the input images, with fixed $\ell_\infty = \frac{8}{255}$, before computing the gradients of the model parameters and updating them. The classification error on the original ImageNet dataset achieved by each model after adversarial fine-tuning is 51.7% and 49.8%, for both models respectively. Figure 1 (right) shows in dashed lines the classification error when these models are evaluated against an adversarial attacker using different $\ell_\infty$ constraints. As the $\ell_\infty$ increases, the benefit of adversarial fine-tuning to preserve accuracy is shown in both cases. Once again, both models report similar trends, and the V-MoE model shows better robustness for a wide range of $\ell_\infty$ values.

---

[1]Token refers to an element in the sequence input, which is obtained by projecting input image patches [7]

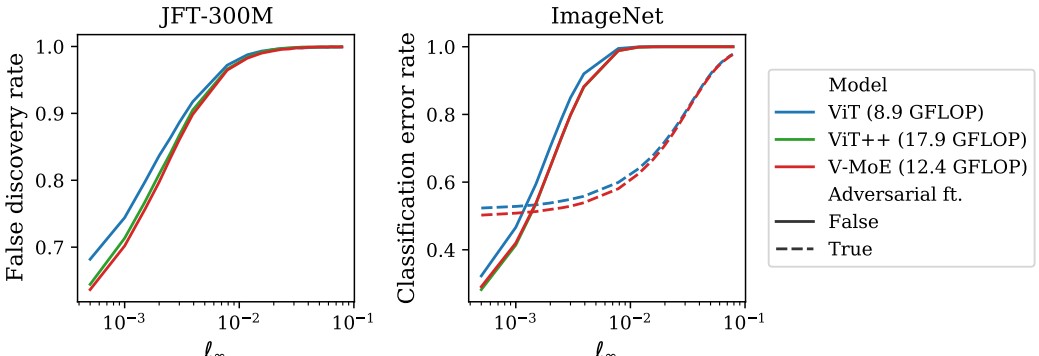

Figure 1: False discovery rate on JFT-300M (left) and classification error rate on ImageNet (right) as a function of the $\ell_\infty$ used in the adversarial attacks. Dashed lines depict models fine-tuned with PGD adversarial training. Although the V-MoE model contains several sparse MoE layers making discrete choices on their respective inputs, it shows lower error under adversarial attacks than the base ViT model. A much bigger and slower ViT model is needed to roughly match the V-MoE.

### 4.3 Effect of the adversarial attacks on the selected experts

As described in section 3.1, in the region close to the decision boundary of the router, if two experts have very different outputs, the Lipschitz constant of the MoE model could be much higher than that of a similar dense model. If the index of the selected experts changes significantly, but the model still shows a reasonably high accuracy under adversarial attacks (compared to a dense model), this would suggest that the outputs of the two selected sets of experts do not differ much.

Figure 2 shows the rate of changes in the router as a function of the $\ell_\infty$ used in the adversarial attack, on the different layers of the V-MoE-B/32 model that have a MoE layer. For each token processed by the model, we compute the intersection-over-union (IoU) of the selected set of experts before and after the adversarial attack. We average the IoU across all processed tokens and define the rate of routing changes as the complement of the average IoU.

In dashed lines we report the rate of changes of a V-MoE model fine-tuned using PGD adversarial training. Not only the model has a better accuracy against adversarial attacks when the $\ell_\infty$ increases, as reported in figure 1, but the rate of routing changes is also generally lower across all layers.

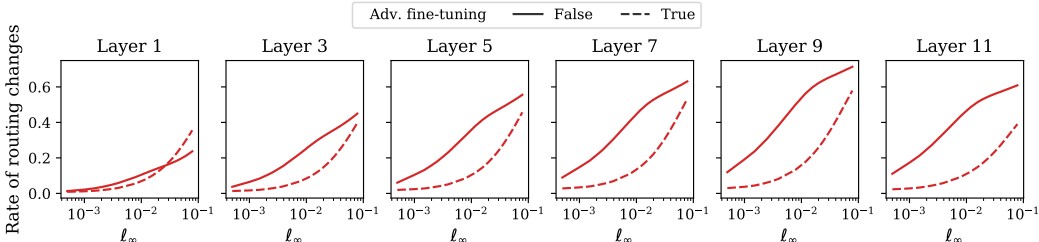

Figure 2: Rate of changes in the selection of experts as a function of the $\ell_\infty$ used in the adversarial attacks on ImageNet against a V-MoE model. The dashed line depicts a V-MoE model fine-tuned with PGD adversarial training. A significant fraction of experts change *in each MoE layer*. Given that figure 1 shows the V-MoE model keeping its advantage over the dense ViT, we hypothesize that the output of different experts for inputs close to the decision boundary of the router is similar.

Despite the fact that a significant fraction of choices change *on each layer* as the $\ell_\infty$ increases, figure 1 shows that the V-MoE model still keeps its advantage against adversarial attacks over the ViT model. This suggests that in the regions close to a decision boundary of the router, the two corresponding experts have a similar output, hence preventing the full V-MoE model from being less robust in practice. Conversely, the fact that the V-MoE model has a better base quality than the ViT counterpart, suggests that the experts are not equivalent for regions far away from the decision boundary, otherwise it would be reduce to a dense model.

## 4.4 Attacking the router's auxiliary losses

Sparse MoE models usually employ auxiliary losses to balance the load among all experts. In particular, in the implementation of V-MoEs, if the load of the experts is highly unbalanced, the experts receiving significantly more tokens than the average could ignore all tokens that exceed the expert's capacity, potentially leading to a significantly worse performance. The question is whether an adversarial attacker can exploit this property in practice. Figure 3 (left) shows the false discovery rate on JFT-300M, for a V-MoE model when the router's auxiliary losses are maximized in the adversarial attack, together with the corresponding cross-entropy loss. We use the same weight for the auxiliary losses as the one used to train the models. The figure shows that attacking the auxiliary loss does not offer any significant advantage for the attacker. The results are analogous in terms of the classification error on ImageNet (not shown here in interest of space).

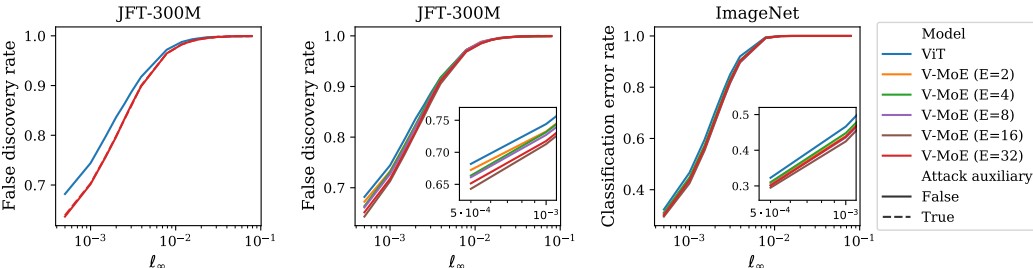

Figure 3: **(Left)** False discovery rate on JFT-300M for a ViT-B/32 and a V-MoE-B/32 models (with 32 experts). Dashed lines represent the results when the attacker targets both the cross-entropy and the auxiliary losses used by the V-MoE routers to balance the load among experts in each layer. The auxiliary losses of V-MoE do not present a disadvantage against an adversarial attacker. **(Center and Right)** False discovery rate on JFT-300M and classification error rate on ImageNet for a ViT-B/32 and several V-MoE-B/32 models (with increasing number of total experts). On JFT-300M the quality of the model against adversarial attacks improves as more experts are used. After fine-tuning on ImageNet, the number of experts becomes increasingly redundant.

## 4.5 Increasing the model size by increasing the total number of experts

In section 1 we asked if, given that the lower bound on the Lipschitz constant given by Bubeck and Sellke [3] is agnostic to the computational cost of the function, could we make a model more robust by increasing its model size without increasing the total cost? Here we measure how much increasing the total number of experts in a V-MoE model helps against adversarial attacks. Notice that increasing the total number of experts $E$ does not make the V-MoE model more expensive. Figure 3 (center and right) shows the false discovery rate on JFT-300M and the classification error on ImageNet for an increasing number of experts. All V-MoE models select $K = 2$ experts for each token.

On the one hand, increasing the total number of experts improves the robustness on JFT-300M up to $E = 16$ experts. The curves for $E = 16$ and $E = 32$ are highly overlapping, thus any difference is most likely due to noise in the training and fine-tuning process. On the other hand, when the V-MoEs are fine-tuned on ImageNet, all models with more than two experts achieve roughly the same accuracy under adversarial attacks.

This shows that, although the results presented in section 3 showing better robustness of MoEs require some assumptions, the conclusions hold to some extent in real scenarios. Increasing the number of parameters by growing the number of experts is an effective way of improving the model's robustness.

## 4.6 Robustness against AutoPGD attacks

We conducted additional experiments using a more sophisticated adversarial attack, AutoPGD [5], which selects the step size to use in each update of the attack. We also increased the number of steps performed in the attack, using $\tau = 100$ steps. Figure 4a shows the results achieved by two dense (ViT-B/32 and ViT-B++/32) models and a sparse model (V-MoE-B/32). Although AutoPGD is slightly more effective as an adversarial attack against all methods (for example, with $\ell_\infty = 10^{-3}$ the false discovery rate on JFT-300M of ViT-B/32 is 0.744 using PGD and 0.757 using AutoPGD), the trend of all models is identical to that represented in figure 1.

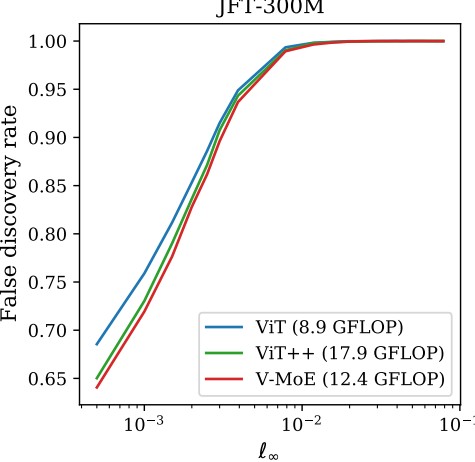

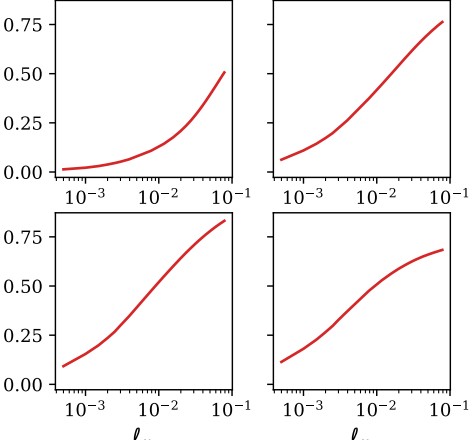

(a) False discovery rate on JFT-300M (left) and classification error rate on ImageNet (right) as a function of the $\ell_\infty$, using AutoPGD and yielding the same conclusions as the simpler PGD method.

(b) Rate of changes in the selection of experts as a function of the $\ell_\infty$ used in the adversarial attacks on ImageNet against a V-MoE model.

Figure 4b shows the rate of routing changes in the different MoE layers in the V-MoE-B/32 model. This figure is analogous to figure 2 which shows the results for the standard PGD attack. Compared to it, AutoPGD is able to change a higher fraction of the selected experts. For example, the maximum fraction of changes in Layer 1 using PGD was near 0.2, while AutoPGD increases it up to around 0.5.

AutoPGD offers the same conclusion as the PGD attacks: models using sparse MoE layers offer better robustness against adversarial attacks (per GFLOP) than dense models, despite the fact that the router itself can be quite sensible to these attacks.

## 5    Conclusion

In this work we analyzed the adversarial robustness of MoEs showing their advantage over dense models, with more experts leading to better robustness, both theoretically and empirically. We showed how the properties of the data and its routing plays and important role in learning robust MoEs. While there is some evidence that routing learned by MoEs in practice display some clustering of the features in some layers of the model (see Figure 7 in  Riquelme et al. [22]), it is currently not explicitly encouraged during training. Hence developing smarter routing strategies that take data geometry into account can be an interesting direction of future work. Currently our analysis is limited to linear models, extending this to general models and deriving the dependency of optimal routing on them is another promising research direction.

We have also shown that, for inputs that weigh two experts similarly, if the two expert values are very different, then the MoEs can suffer from higher Lipschitz constant. However, for models trained in practice, we saw their predictions to be relatively stable, despite significant changes in choice of experts, highlighting potential redundancy of learned experts. However too much redundancy, with all experts learning similar functions is a waste of capacity and can affect model performance. Hence it is an interesting research problem to balance robustness and accuracy of MoEs by controlling the redundancy of experts.

## Acknowledgments and Disclosure of Funding

All authors would like to thank Sven Gowal for sharing their AutoPGD attack codebase. We also thank Neil Houlsby, Basil Mustafa, and André Susano Pinto for providing insightful feedback while working on this project.

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
