# A Proofs

*Proof of lemma 1.* We have by chain rule

$$\nabla_{\boldsymbol{x}} f(\boldsymbol{x}) = \sum_{i=1}^{k} p_i(\boldsymbol{x}) \nabla f_i(\boldsymbol{x}) + \sum_{i=1}^{k} \nabla p_i(\boldsymbol{x}) f_i(\boldsymbol{x}).$$

Hence,

$$\|\nabla_{\boldsymbol{x}} f(\boldsymbol{x})\| \le \sum_{i=1}^{k} p_i(\boldsymbol{x}) L_{f_i} + \|\sum_{i=1}^{k} \nabla p_i(\boldsymbol{x}) f_i(\boldsymbol{x})\|,$$

where

$$
\begin{aligned}
\nabla p_i(\boldsymbol{x}) &= \nabla \frac{\exp(\langle \boldsymbol{s}_i, \boldsymbol{x} \rangle)}{\sum_j \exp(\langle \boldsymbol{s}_j, \boldsymbol{x} \rangle)} \\
&= \frac{\sum_j \exp(\langle \boldsymbol{s}_j, \boldsymbol{x} \rangle) \cdot \exp(\langle \boldsymbol{s}_i, \boldsymbol{x} \rangle) \boldsymbol{s}_i - \exp(\langle \boldsymbol{s}_i, \boldsymbol{x} \rangle) \cdot \sum_j \exp(\langle \boldsymbol{s}_j, \boldsymbol{x} \rangle) \boldsymbol{s}_j}{\sum_j \exp(\langle \boldsymbol{s}_j, \boldsymbol{x} \rangle) \cdot \sum_j \exp(\langle \boldsymbol{s}_j, \boldsymbol{x} \rangle)} \\
&= \frac{\exp(\langle \boldsymbol{s}_i, \boldsymbol{x} \rangle) \boldsymbol{s}_i}{\sum_j \exp(\langle \boldsymbol{s}_j, \boldsymbol{x} \rangle)} - p_i \frac{\sum_j \exp(\langle \boldsymbol{s}_j, \boldsymbol{x} \rangle) \boldsymbol{s}_j}{\sum_j \exp(\langle \boldsymbol{s}_j, \boldsymbol{x} \rangle)} \\
&= p_i \boldsymbol{s}_i - p_i \sum_j p_j \boldsymbol{s}_j.
\end{aligned}
$$

Let $\bar{\boldsymbol{s}}(\boldsymbol{x}) = \sum_j p_j(\boldsymbol{x}) \boldsymbol{s}_j$. Substituting this in the above equation gives us

$$
\begin{aligned}
\|\nabla_{\boldsymbol{x}} f(\boldsymbol{x})\| &\le \sum_{i=1}^{k} p_i(\boldsymbol{x}) L_{f_i} + \|\sum_{i=1}^{k} p_i(\boldsymbol{x}) f_i(\boldsymbol{x}) (\boldsymbol{s}_i - \bar{\boldsymbol{s}}(\boldsymbol{x}))\| \\
&\le \max_i \{L_{f_i}\} + \max_x \|\sum_{i=1}^{E} p_i(\boldsymbol{x}) f_i(\boldsymbol{x}) (\boldsymbol{s}_i - \bar{\boldsymbol{s}}(\boldsymbol{x}))\|.
\end{aligned}
$$

The last inequality follows from Holder's inequality and $\sum_{i=1}^{k} p_i(\boldsymbol{x}) = 1$. $\qquad\square$

*Proof of theorem 1.* Since $\boldsymbol{U}_i$ are orthogonal to each other, we can decompose $\|\boldsymbol{w}^*\|$ as follows.

$$\|\boldsymbol{w}^*\|^2 = \|(\boldsymbol{X}^\top \boldsymbol{X})^\dagger \boldsymbol{X}^\top \boldsymbol{y}\|^2 = \|(\sum_i \boldsymbol{U}_i \boldsymbol{U}_i^\top)(\boldsymbol{X}^\top \boldsymbol{X})^\dagger \boldsymbol{X}^\top \boldsymbol{y}\|^2 = \sum_i \|\boldsymbol{U}_i \boldsymbol{U}_i^\top (\boldsymbol{X}^\top \boldsymbol{X})^\dagger \boldsymbol{X}^\top \boldsymbol{y}\|^2. \tag{10}$$

From definition 1 we have, for any unit norm vector $\boldsymbol{z}$,

$$
\begin{aligned}
\epsilon_1 &\ge \|\boldsymbol{U}_i \boldsymbol{U}_i^\top (\boldsymbol{X}^\top \boldsymbol{X})^\dagger - (\boldsymbol{X}_i^\top \boldsymbol{X}_i)^\dagger\|_2 \\
&\ge \|\boldsymbol{U}_i \boldsymbol{U}_i^\top (\boldsymbol{X}^\top \boldsymbol{X})^\dagger \boldsymbol{z} - (\boldsymbol{X}_i^\top \boldsymbol{X}_i)^\dagger \boldsymbol{z}\| \\
&\ge \left| \|\boldsymbol{U}_i \boldsymbol{U}_i^\top (\boldsymbol{X}^\top \boldsymbol{X})^\dagger \boldsymbol{z}\| - \|(\boldsymbol{X}_i^\top \boldsymbol{X}_i)^\dagger \boldsymbol{z}\| \right|
\end{aligned}
$$

The last step follows from triangle inequality. This implies

$$\|(\boldsymbol{X}_i^\top \boldsymbol{X}_i)^\dagger \boldsymbol{z}\| \le \|\boldsymbol{U}_i \boldsymbol{U}_i^\top (\boldsymbol{X}^\top \boldsymbol{X})^\dagger \boldsymbol{z}\| + \epsilon_1.$$

Hence, by setting $\boldsymbol{z} = \boldsymbol{X}^\top \boldsymbol{y} / \|\boldsymbol{X}^\top \boldsymbol{y}\|$, we see that

$$\|\boldsymbol{U}_i \boldsymbol{U}_i^\top (\boldsymbol{X}^\top \boldsymbol{X})^\dagger \boldsymbol{X}^\top \boldsymbol{y}\| \ge \|(\boldsymbol{X}_i^\top \boldsymbol{X}_i)^\dagger \boldsymbol{X}^\top \boldsymbol{y}\| - \epsilon_1 \|\boldsymbol{X}^\top \boldsymbol{y}\|.$$

Recall that $\boldsymbol{X}^\top \boldsymbol{y} = \sum_{j=1}^E \boldsymbol{X}_j^\top \boldsymbol{y}_j$. Then using triangle inequality, and the above inequality, we get,

$$
\begin{aligned}
\|\boldsymbol{w}_i^*\| &= \|(\boldsymbol{X}_i^\top \boldsymbol{X}_i)^\dagger \boldsymbol{X}_i^\top \boldsymbol{y}_i\| \\
&\leq \|(\boldsymbol{X}_i^\top \boldsymbol{X}_i)^\dagger \boldsymbol{X}^\top \boldsymbol{y}\| + \sum_{j \neq i} \|(\boldsymbol{X}_i^\top \boldsymbol{X}_i)^\dagger \boldsymbol{X}_j^\top \boldsymbol{y}_j\| \\
&\leq \|\boldsymbol{U}_i \boldsymbol{U}_i^\top (\boldsymbol{X}^\top \boldsymbol{X})^\dagger \boldsymbol{X}^\top \boldsymbol{y}\| + \epsilon_1 \|\boldsymbol{X}^\top \boldsymbol{y}\| + \sum_{j \neq i} \|(\boldsymbol{X}_i^\top \boldsymbol{X}_i)^\dagger \boldsymbol{X}_j^\top \boldsymbol{y}_j\| \\
&\leq \|\boldsymbol{U}_i \boldsymbol{U}_i^\top (\boldsymbol{X}^\top \boldsymbol{X})^\dagger \boldsymbol{X}^\top \boldsymbol{y}\| + \epsilon_1 \|\boldsymbol{X}^\top \boldsymbol{y}\| + \epsilon_2 \sum_{j \neq i} \|\boldsymbol{X}_j^\top \boldsymbol{y}_j\|.
\end{aligned}
$$

The last step follows from definition 2. Thus,

$$
\|\boldsymbol{U}_i \boldsymbol{U}_i^\top (\boldsymbol{X}^\top \boldsymbol{X})^\dagger \boldsymbol{X}^\top \boldsymbol{y}\| \geq \|\boldsymbol{w}_i^*\| - \epsilon_1 \|\boldsymbol{X}^\top \boldsymbol{y}\| - \epsilon_2 \sum_{j \neq i} \|\boldsymbol{X}_j^\top \boldsymbol{y}_j\|.
$$

Substituting this in equation 10 gives us the result.

$$
\|\boldsymbol{w}^*\|^2 \geq \sum_i (\lfloor \|\boldsymbol{w}_i^*\| - \epsilon_1 \|\boldsymbol{X}^\top \boldsymbol{y}\| - \epsilon_2 \sum_{j \neq i} \|\boldsymbol{X}_j^\top \boldsymbol{y}_j\| \rfloor_+)^2.
$$

$\square$

# B Experiments details

## B.1 Architectures

Table 1 contains the details of the architectures used in section 4. All V-MoE models have replaced one in every two feedforward layers in the original ViT-B/32 architecture, with a MoE of feedforward layers, and select only $K = 2$ experts out of the $E$ available experts. All models were pre-trained on JFT-300M, at a resolution of $224 \times 224$ pixels, for a total number of 7 epochs, using the same batch size and optimizer settings as used by Dosovitskiy et al. [7] and Riquelme et al. [22] (see details section 4). We used the B/32 variants of ViT and V-MoE since these are the "base" configurations suggested in the respective papers.

In order to match the quality of a dense ViT and a sparse V-MoE model, we also trained a bigger version of ViT-B/32, which we refer to as ViT-B++/32. The values for the number of layers, number of attention heads, embedding dimension and the hidden dimension of the FFN, are simply an interpolation between the ViT-B/32 and the ViT-L/32 corresponding values.

We implemented all models, training and evaluation code using JAX[2] and FLAX[3]. Although we cannot release the models used in the experiments, since they are pre-trained on proprietary data, the code used in all of them is available at `http://github.com/google-research/vmoe`.

## B.2 Compute resources

We use TPUv3 to train and evaluate our models. In particular, we used 32 TPUv3 cores for pre-training and fine-tuning the models. In most of the evaluation experiments against adversarial attacks we used 8 TPUv3 cores, since the compute needed to perform the evaluation against adversarial attack is much lower. The total training cost (in terms of total training runtime and FLOPs) of ViT-B/32 and V-MoE-B/32 can be found in [22].

ViT-B/32 is pre-trained for a total of 27.6 TPUv3-core-days, performing 56.1 ExaFLOPs. V-MoE-B/32 (with 32 experts) is trained for a total of 54.9 TPUv3-core-days and performs a total of 76.1 ExaFLOPs. Finally, the larger dense ViT-B++/32 is trained for a total of 78.3 TPUv3-core-days and performs 113.4 ExaFLOPs.

The GFLOPs are computed automatically by JAX. For instance, to compute the GFLOPs used for each individual image during the evaluation of the models, the code snippet in listing 1 is used.

---

[2] `https://github.com/google/jax`
[3] `https://github.com/google/flax`

Table 1: Architecture details of all the models used in the experiments. We specify here the number of layers in the Transformer encoder, the number of heads used in the multi-head attention, the embedding dimension (i.e. token size), the hidden size in the feedforward (FFN) layers and experts, the selected and total number of experts (when applicable), the active/total number of parameters, and the cost of evaluation per image (at a resolution of $224 \times 224$ pixels). Precision-at-1 on the pre-training dataset (JFT-300M) and classification error rate on ImageNet (ILSVRC2012) are also reported.

| Name | ViT-B/32 | ViT-B++/32 | V-MoE-B/32 | | | | |
| --- | --- | --- | --- | --- | --- | --- | --- |
| | | | E=2 | E=4 | E=8 | E=16 | E=32 |
| Layers | 12 | 18 | | | 12 | | |
| Heads | 12 | 14 | | | 12 | | |
| Embedding dim. | 768 | 896 | | | 768 | | |
| FFN dim. | 3072 | 3584 | | | 3072 | | |
| Selected/Total experts | — | — | 2/2 | 2/4 | 2/8 | 2/16 | 2/32 |
| Active parameters (M) | 102.1 | 193.6 | 130.5 | 130.5 | 130.5 | 130.5 | 130.6 |
| Total parameters (M) | 102.1 | 193.6 | 130.5 | 187.1 | 300.5 | 527.2 | 980.6 |
| Eval. GFLOPs/image | 8.9 | 17.9 | 12.1 | 12.2 | 12.2 | 12.3 | 12.4 |
| JFT-300M P@1 (%) | 39.3 | 42.9 | 40.5 | 41.4 | 42.7 | 43.6 | 43.5 |
| ILSVRC2012 error (%) | 19.3 | 17.2 | 18.9 | 18.7 | 18.0 | 17.8 | 17.8 |

Listing 1: Code snippet used to compute the evaluation GFLOPs per image.

```
1 client = jax.lib.xla_bridge.get_backend()
2 # eval_step_fn is the function called to evaluate one batch.
3 m = jax.xla_computation(eval_step_fn)(...).as_hlo_module()
4 analysis = jax.lib.xla_client._xla.hlo_module_cost_analysis(client, m)
5 eval_gflops_per_image = analysis['flops'] / 10**9 / batch_size
```

## C  Adversarial examples

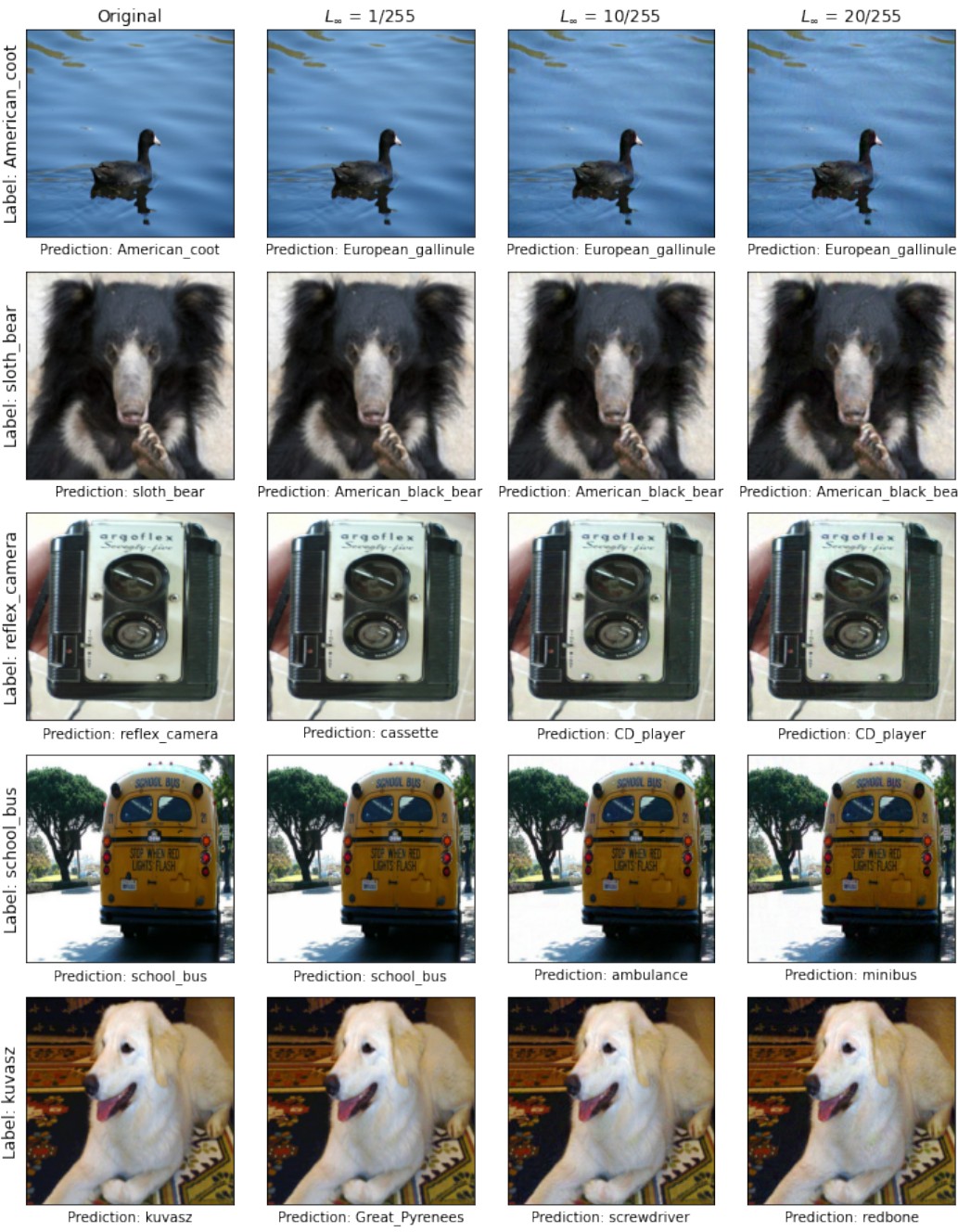

Figure 5: ImageNet adversarial examples generated when attacking a V-MoE-B/32, for different values of $L_{\mathrm{inf}}$. Although the differences are hard to spot (better seen in a monitor, zooming in the images), they are sufficient to change the prediction of the model.

## D  A complementary result to theorem 1

We now present our complementary result. In essence, in a regression setting, our result tries to capture how different the sub-problems tackled by the experts should be in order to get improved robustness compared with a dense approach.

In what follows, we use the shortcuts $\boldsymbol{Z} = \boldsymbol{X}^\top \boldsymbol{X} \in \mathbb{R}^{D \times D}$ and $\boldsymbol{Z}_i = \boldsymbol{X}_i^\top \boldsymbol{X}_i \in \mathbb{R}^{D \times D}$ for $i \in [E]$. Also, we recall that each $\boldsymbol{X}_i$ is in $\mathbb{R}^{N_i \times D}$ with $\sum_{i=1}^{E} N_i = N$.

**Theorem 2.** *Let $\boldsymbol{w}^*$ be the minimizer of equation 8 and $\{\boldsymbol{w}_i^*\}_{i \in [E]}$ be the minimizers of equation 9. For all $i \in [E]$, let us assume that there exist $\beta_i \in \mathbb{R}^D$ and $\boldsymbol{r}_i \in \mathbb{R}^{N_i}$ such that we can write*

$$\boldsymbol{y}_i = \boldsymbol{X}_i \beta_i + \boldsymbol{r}_i$$

*where the vectors $\boldsymbol{r}_i$'s are residual terms that can for instance account for non-linear effects in $\boldsymbol{X}_i$ and/or some stochastic noise. Further assume the $\{\boldsymbol{Z}_i\}_{i \in [E]}$ to be invertible and define*

$$\eta_i = \boldsymbol{Z}_i^{-1} \boldsymbol{X}_i^\top \boldsymbol{r}_i \quad and \quad \eta = \boldsymbol{Z}^{-1} \boldsymbol{X}^\top \boldsymbol{r} \ with \ \boldsymbol{r} = [\boldsymbol{r}_1, \dots, \boldsymbol{r}_E] \in \mathbb{R}^N.$$

*Let us denote the (normalized) difference between any two vectors $(\beta_i, \beta_j)$ as $\Delta_{ij} \in \mathbb{R}^D$, that is*

$$for \ any \ i \neq j, \ \Delta_{ij} = \boldsymbol{Z}^{-1} \boldsymbol{Z}_j (\beta_i - \beta_j),$$

*and consider the cumulative difference with respect to the vector $\beta_i$ as*

$$for \ any \ i \in [E], \ \Delta_i = \sum_{j \neq i} \Delta_{ij} \in \mathbb{R}^D.$$

*Consider $\gamma \in (0, 1]$. If the structures of the underlying sub-problems on $\{(\boldsymbol{X}_i, \boldsymbol{y}_i)\}_{i \in [E]}$ differ sufficiently from each other in the following sense*

$$for \ all \ i \in [E], \ \|\Delta_i\| \geq \|\beta_i + \eta\| + \frac{1}{\sqrt{\gamma}} \|\beta_i + \eta_i\|,$$

*then it holds that*

$$\max_{i \in [E]} \|\boldsymbol{w}_i^*\|^2 \leq \gamma \|\boldsymbol{w}^*\|^2.$$

*In other words, the experts have smaller Lipschitz constants than the dense model by a factor $\gamma$. As a corollary, if we can control the residual terms $\boldsymbol{r}_i$'s such that*

$$\|\eta\| \leq \min_{i \in [E]} \|\beta_i\| \ and \ for \ all \ i \in [E], \ \|\eta_i\| \leq \|\beta_i\|,$$

*then the same conclusion is implied by the simpler sufficient condition*

$$\min_{i \in [E]} \left\{ \|\Delta_i\| - 4/\sqrt{\gamma} \cdot \|\beta_i\| \right\} \geq 0.$$

*Proof of theorem 2.* As a first comment, since we assume that all the $\boldsymbol{Z}_i$'s are invertible, then so is $\boldsymbol{Z} = \sum_{i=1}^{E} \boldsymbol{Z}_i$. We readily have for $i \in [E]$

$$\boldsymbol{w}_i^* = \boldsymbol{Z}_i^{-1} \boldsymbol{X}_i^\top \boldsymbol{y}_i = \beta_i + \boldsymbol{Z}_i^{-1} \boldsymbol{X}_i^\top \boldsymbol{r}_i = \beta_i + \eta_i.$$

Moreover, for any $i \in [E]$, we have

$$
\begin{aligned}
\boldsymbol{w}^* &= \boldsymbol{Z}^{-1} \boldsymbol{X}^\top \boldsymbol{y} \\
&= \boldsymbol{Z}^{-1} \sum_{j=1}^{E} \boldsymbol{X}_j^\top \boldsymbol{y}_j \\
&= \boldsymbol{Z}^{-1} \sum_{j=1}^{E} (\boldsymbol{Z}_j \beta_j + \boldsymbol{X}_j^\top \boldsymbol{r}_j) \\
&= \left\{ \boldsymbol{Z}^{-1} \sum_{j=1}^{E} \boldsymbol{Z}_j (\beta_i - \boldsymbol{Z}_j^{-1} \boldsymbol{Z} \Delta_{ij}) \right\} + \eta \\
&= \boldsymbol{Z}^{-1} \left\{ \sum_{j=1}^{E} \boldsymbol{Z}_j \right\} \beta_i - \sum_{j=1}^{E} \Delta_{ij} + \eta \\
&= \beta_i - \Delta_i + \eta \\
&= \boldsymbol{w}_i^* - \Delta_i + \eta - \eta_i.
\end{aligned}
$$

To summarize, for any $i \in [E]$, it therefore holds

$$\|\boldsymbol{w}^* - \boldsymbol{w}_i^*\| = \|\eta - \eta_i - \Delta_i\|.$$

Let us consider $\gamma \in (0, 1]$. We can observe that

$$\gamma\|\boldsymbol{w}^*\|^2 - \|\boldsymbol{w}_i^*\|^2 = \gamma\|\boldsymbol{w}^* - \boldsymbol{w}_i^*\|^2 - (1 + \gamma)\|\boldsymbol{w}_i^*\|^2 + 2\gamma(\boldsymbol{w}^*)^\top \boldsymbol{w}_i^*.$$

We develop each term individually:

$$\gamma\|\boldsymbol{w}^* - \boldsymbol{w}_i^*\|^2 = \gamma\|\eta - \eta_i\|^2 + \gamma\|\Delta_i\|^2 - 2\gamma\Delta_i^\top(\eta - \eta_i) \quad \textbf{(A)}$$

with

$$-(1 + \gamma)\|\boldsymbol{w}_i^*\|^2 = -(1 + \gamma)\|\beta_i + \eta_i\|^2 \quad \textbf{(B)}$$

and using $\boldsymbol{w}^* = \beta_i - \Delta_i + \eta = \beta_i + \eta_i - \Delta_i + \eta - \eta_i$, we get

$$2\gamma(\boldsymbol{w}^*)^\top \boldsymbol{w}_i^* = 2\gamma\|\beta_i + \eta_i\|^2 - 2\gamma\Delta_i^\top(\beta_i + \eta_i) + 2\gamma(\beta_i + \eta_i)^\top(\eta - \eta_i). \quad \textbf{(C)}$$

Gathering **(A)+(B)+(C)**, we obtain the expression

$$\begin{aligned}
\gamma\|\boldsymbol{w}^*\|^2 - \|\boldsymbol{w}_i^*\|^2 &= \gamma\|\boldsymbol{w}^* - \boldsymbol{w}_i^*\|^2 - (1 + \gamma)\|\boldsymbol{w}_i^*\|^2 + 2\gamma(\boldsymbol{w}^*)^\top \boldsymbol{w}_i^* \\
&= \gamma\|\Delta_i\|^2 - 2\gamma\Delta_i^\top(\beta_i + \eta) + \gamma\|\beta_i + \eta\|^2 - \|\beta_i + \eta_i\|^2 = \phi(\Delta_i).
\end{aligned}$$

We want to understand the conditions on $\Delta_i$ to guarantee that

$$\gamma\|\boldsymbol{w}^*\|^2 - \|\boldsymbol{w}_i^*\|^2 = \phi(\Delta_i) \geq 0.$$

To find a sufficient condition, we lower bound the quadratic form $\phi(\Delta_i)$ as follows

$$\begin{aligned}
\text{For any } \Delta_i \in \mathbb{R}^D, \ \phi(\Delta_i) &\geq \gamma\|\Delta_i\|^2 - 2\gamma\|\Delta_i\|\|\beta_i + \eta\| + \gamma\|\beta_i + \eta\|^2 - \|\beta_i + \eta_i\|^2 \\
&= \gamma\left(\|\Delta_i\| - \text{root}_+\right) \cdot \left(\|\Delta_i\| - \text{root}_-\right)
\end{aligned}$$

with

$$\text{root}_+ = \|\beta_i + \eta\| + \frac{1}{\sqrt{\gamma}}\|\beta_i + \eta_i\| \quad \text{and} \quad \text{root}_- = \|\beta_i + \eta\| - \frac{1}{\sqrt{\gamma}}\|\beta_i + \eta_i\|.$$

The lower bound is a standard polynomial of degree 2 and it is non-negative beyond its positive root

$$\|\Delta_i\| \geq \|\beta_i + \eta\| + \frac{1}{\sqrt{\gamma}}\|\beta_i + \eta_i\|.$$

We apply the exact same rationale for all $i \in [E]$ and take the minimum of the conditions to make them hold simultaneously.

Since $\gamma \in (0, 1]$ and $1/\sqrt{\gamma} \geq 1$, if we further assume that the residual terms are sufficiently small

- $\|\eta\| \leq \min_{i \in [E]} \|\beta_i\|$

- for all $i \in [E]$, $\|\eta_i\| \leq \|\beta_i\|$

then we get the simpler sufficient condition after using the triangle inequality

$$\|\Delta_i\| \geq \frac{4}{\sqrt{\gamma}}\|\beta_i\| \geq \|\beta_i + \eta\| + \frac{1}{\sqrt{\gamma}}\|\beta_i + \eta_i\|.$$

$\square$

# E   Auxiliary losses for MoEs

In this section we present the Auxiliary losses used for training MoEs and attack objectives used for our experiments in section 4. Auxiliary losses are used to ensure the data is routed in a balanced way across different experts. In particular we use the losses from Riquelme et al. [22] which are defined below.

Recall that the router weights for a given token $\boldsymbol{x}$ and a given expert $j$, with routing parameters $\{\boldsymbol{s}_j\}_{j=1}^E$, were defined in equation 6 as:

$$p_j(\boldsymbol{x}) = \frac{\exp(\langle \boldsymbol{s}_j, \boldsymbol{x}\rangle)}{\sum_{j'=1}^E \exp(\langle \boldsymbol{s}_j', \boldsymbol{x}\rangle)}$$

In practice, we use a noisy version of this router:

$$p_j(\boldsymbol{x}) = \frac{\exp(\langle \boldsymbol{s}_j, \boldsymbol{x}\rangle + \epsilon_j)}{\sum_{j'=1}^E \exp(\langle \boldsymbol{s}_j', \boldsymbol{x}\rangle + \epsilon_{j'})}$$

with $\epsilon_j \sim \mathcal{N}(\mu = 0, \sigma = \frac{1}{E})$. The quantity $\langle \boldsymbol{s}_j, \boldsymbol{x}\rangle$ is the routing *logit* for a given token and expert $j$.

**Importance Loss.** The importance of expert $j$ for a batch of tokens $\boldsymbol{X}$ is simply defined as the sum of the router weights assigned to that expert over the tokens in the batch:

$$\mathrm{Imp}_j(\boldsymbol{X}) = \sum_i p_j(\boldsymbol{x}_i) \tag{11}$$

where $\boldsymbol{X} = [\boldsymbol{x}_1, \ldots, \boldsymbol{x}_n]^\top$ and $p_j(\boldsymbol{x}_i)$ is the routing weight assigned to the $j$-th expert for the $i$-th token.

We use the squared coefficient of variation of the importance distribution over experts, $\mathrm{Imp}(\boldsymbol{X}) := \{\mathrm{Imp}_j(\boldsymbol{X})\}_{j=1}^E$:

$$\mathcal{L}_{\mathrm{Imp}}(\boldsymbol{X}) = \left(\frac{\mathrm{std}(\mathrm{Imp}(\boldsymbol{X}))}{\mathrm{mean}(\mathrm{Imp}(\boldsymbol{X}))}\right)^2 \tag{12}$$

**Load Loss.** For a given token, the load loss is based on the probability that the noisy logit of expert $j$ is among the top-$k$ logits when a new Gaussian sample is drawn. Let's define the $k$-th maximum noisy logit for a given token $\boldsymbol{x}$ as:

$$\tau_k(\boldsymbol{x}) = k\text{–}\max_j \langle \boldsymbol{s}_j, \boldsymbol{x}\rangle + \epsilon_j \tag{13}$$

Then, the probability that the noisy logits of expert $j$ are above the threshold if the noise is resampled, is given by the expression:

$$\pi_j(\boldsymbol{x}) = \mathbb{P}(\langle \boldsymbol{s}_j, \boldsymbol{x}\rangle + \epsilon \geq \tau_k(\boldsymbol{x})) \tag{14}$$

Analogous to the importance, the load of expert $j$ is defined as the sum over the tokens in a given batch:

$$\mathrm{Load}_j(\boldsymbol{X}) = \sum_i \pi_j(\boldsymbol{x}_i) \tag{15}$$

And the load loss corresponds to the squared coefficient of variation of the load distribution, with $\mathrm{Load}(\boldsymbol{X}) := \{\mathrm{Load}_j(\boldsymbol{X})\}_{j=1}^E$:

$$\mathcal{L}_{\mathrm{Load}}(\boldsymbol{X}) = \left(\frac{\mathrm{std}(\mathrm{Load}(\boldsymbol{X}))}{\mathrm{mean}(\mathrm{Load}(\boldsymbol{X}))}\right)^2 \tag{16}$$

**Final Loss.** The final loss used for training the models includes the classification loss (i.e. cross-entropy) and both auxiliary losses with weights 0.005:

$$\mathcal{L} = \mathcal{L}_{\mathrm{classification}} + 0.005\,\mathcal{L}_{\mathrm{Imp}} + 0.005\,\mathcal{L}_{\mathrm{Load}} \tag{17}$$

# F  Experiment Metrics

**Precision at 1 and False discovery rate**  In multi-label datasets (such as JFT), the traditional accuracy metric used for single-label classification is ill-defined. Usually, *Precision at $k$*, *Recall at $k$*, and other more general metrics are used in this setting. In particular, we use the *Precision at 1*.

Let $\boldsymbol{Y} = [\boldsymbol{y}_1, \ldots, \boldsymbol{y}_N]^\top, \boldsymbol{y}_i \in \mathbb{R}^C$ be the logits output by the model for a set of $N$ examples, with true labels $\hat{\boldsymbol{Y}} = [\hat{\boldsymbol{y}}_1, \ldots, \hat{\boldsymbol{y}}_N]^\top, \hat{\boldsymbol{y}}_i \in \{0, 1\}^C$. *Precision at 1* is defined as:

$$\mathrm{P@1} = \frac{1}{N}\sum_{i=1}^N \delta[\hat{\boldsymbol{y}}_{i, \arg\max_c \boldsymbol{y}_{i,c}} = 1] \tag{18}$$

where $\arg\max_c \boldsymbol{y}_{i,c}$ is essentially the index of the class with the highest logit, for a given example $i$, and $\delta$ is the Kronecker delta function. That is, the non-zero elements of the sum are those examples for which the highest logit corresponds to one of the true labels of such example. The false discovery rate is defined as $1 - P@1$.

Observe than when the examples have a single label, precision at 1 is equivalent to the accuracy, and the false discovery rate is equivalent to the error rate.

**Rate of routing changes**  Given two sets $A$ and $B$, the intersection over union (IoU) is defined as:

$$\text{IoU} = \frac{\text{card}(A \cap B)}{\text{card}(A \cup B)} \tag{19}$$

If the two sets are equal, IoU is 1. If the two sets do not share any element, $\text{IoU} is 0$.

We use this metric in section 4.3 to compare how much the set of selected experts *change* when adversarial attacks are peformed on V-MoE models. Thus, the *rate of routing changes* is defined as $1 - \text{IoU}$.