# OpenReview forum: "On the Adversarial Robustness of Mixture of Experts"
_NeurIPS.cc/2022/Conference — NeurIPS 2022 Accept_

### Official Review · Reviewer_P3QL · 2022-06-18

**Rating:** 6
**Confidence:** 4
**Soundness:** 2 fair
**Presentation:** 3 good
**Contribution:** 3 good

**Summary:**

This paper gives a theoretical analysis to the adversarial robustness of MoEs, it shows MoEs achieve better robustness than the dense models through lower bounding the Lipschitz constant of MoEs by that of the dense models theoretically, and then shows superior robustness of MoEs compared to the dense models experimentally.

**Questions:**

As stated in the previous part, firstly, I think it is better to run the experiments on more attacks, just running it on the PGD attacks is not convincing; secondly, the theory in this paper lower bounds the Lipschitz constant of MoEs by that of the dense models, and then the authors claim that MoEs achieve better robustness than the dense counterparts theoretically, what I concern is that as a global property, can the Lipschitz constant fully describe the robustness of a model? Perhaps a smaller Lipschitz constant means the model will have better robustness in the worst case, i.e. the mass of the data is concentrated on the most steep part of the model.

**Limitations:**

The authors addressed the limitations of this work in the conclusion part.

**Strengths And Weaknesses:**

Strengths:
- This paper lower bounds the Lipschitz constant of MoEs by that of the dense models, showing that MoEs achieve better robustness than the dense models.
- This paper does some experiments to verify the superior robustness of MoEs compared to the dense models and uncovers some intriguing properties of adversarial attacks for MoEs.
- Although the linear regression model and the kernel regression is quite simple, this paper shows some interesting results about the robustness of MoEs.

Weaknesses:
- In the experiments, the paper just uses the PGD adversarial attacks, it is not convincing, as is said in [1], just use PGD attacks is not enough, I suggest to do further experiments on AutoAttack[1] and C&W[2].
- Lipschitz constant is a global property, but adversarial robustness is not a global property, it depends on the data distribution and other factors. So I think the Lipschitz constant is not sufficient to describe the robustness of a classifier.

[1] Francesco Croce and Matthias Hein. Reliable evaluation of adversarial robustness with an ensemble of diverse parameter-free attacks. In ICML, 2020.

[2] Nicholas Carlini and David A. Wagner. Towards evaluating the robustness of neural networks. IEEE Symposium on Security and Privacy (SP), pages 39–57, 2017.

---

> ### Author Response · Authors · 2022-08-02
> **Author response**
>
> We thank the reviewer for their review and finding our analysis and results interesting. Below we address their comments.
>
> **Other attacks:** We agree with the reviewer that including other attacks such as AutoPGD can be useful. However please note that AutoPGD is a significantly more expensive method with [1]  using only 1000 examples from the ImageNet validation set to test AutoPGD. Since there has been no prior work empirically analyzing the robustness of MoEs our goal in experiments has been to test adversarial robustness of large scale MoE models on ImageNet dataset, and we started with PGD as the popular and relatively cheaper choice for adversarial attack. We experimented with a wide range of \epsilon values (1e-3 to 1e-1), and a number of experts (2 to 32). We will provide more results using other attacks in the final version.
>
>
> **Lipschitz constant:**
> We agree with the reviewer about the drawbacks of the relationship between Lipschitz constant  and adversarial robustness. However Lipschitz constant has been the popular proxy used to study and enforce adversarial robustness in the literature. See for instance the papers cited below. Note that we have also presented adversarial robustness results on real large-scale networks, which complements our analysis on the robustness of MoE models.
>
> 1. [Towards Deep Neural Network Architectures Robust to Adversarial Examples](https://arxiv.org/abs/1412.5068)
> 2. [Efficient and accurate estimation of lipschitz constants for deep neural networks](https://dl.acm.org/doi/abs/10.5555/3454287.3455312)
> 3. [Intriguing properties of neural networks](https://arxiv.org/abs/1312.6199)

---

> > ### Comment · Reviewer_P3QL · 2022-08-04
> > **About the relationship between Lipschitz constant and adversarial robustness**
> >
> > Perhaps my statement is not clear before.
> >
> > Surely an upper bound of the Lipschitz constant means an upper bound on the adversarial robustness, the upper bounds in your paper can reflect the robustness of the MoEs in some sense.
> >
> > But I think it is not suitable to compare the adversarial robustness of two models by their Lipschitz constant. For example, let
> >
> > $$f _1(x) =\begin{cases}
> > 0,  & \text{if $x \le 0$} \\\\
> > x, & \text{if $0 < x \le 1$} \\\\
> > 1, & \text{if $x > 1$}
> > \end{cases}$$
> >
> > and let $f _2(x) = 0.5 x$
> >
> > If the probability concentrates in (0,1), then $f _2$ is more robust than $f _1$, but when there is no mass in (0,1), I think $f _1$ may be more robust.
> >
> > I think the main reason cause this result is that the Lipschitz constant is a global property, which means that it has no guarantee on the local part of $\mathcal{X}$, which makes the robustness may vary a lot when considering different distributions.

---

> > > ### Author Response · Authors · 2022-08-05
> > > **Response**
> > >
> > > We agree with the reviewer’s general point, and acknowledge that the Lipschitz constant is just an imperfect proxy for robustness that ignores certain key aspects of ML setups such as the data distribution. In the absence of other more meaningful proxies that are amenable to theoretical analysis, we decided to focus on the Lipschitz constant as it’s a popular one and previous relevant work also analyzed it [1]. Similarly, rather than directly exploring modern deep architectures, we focused on simple linear models. Accordingly, by no means we claim this is the definite angle to theoretically study robustness of deep models, but we hope it’s a step forward. We’ll try to make these observations explicit and clear in the text.
> > >
> > > To show that our theoretical intuition could match what happens in applied setups, we provided extensive empirical results and experiments that seem to agree with our theoretical results.
> > >
> > >
> > >
> > > [1] A universal law of robustness via isoperimetry, Bubeck & Sellke.

---

> > > > ### Comment · Reviewer_P3QL · 2022-08-07
> > > > **changed my score.**
> > > >
> > > > Thanks for your response, you have solved some of my concerns, and I increased my score.

---

### Official Review · Reviewer_wLgw · 2022-06-30

**Rating:** 6
**Confidence:** 4
**Soundness:** 3 good
**Presentation:** 3 good
**Contribution:** 2 fair

**Summary:**

This paper studies the robustness of mixture-of-experts (MOE) models.  The authors study the Lipschitz properties of MOEs, and they study MOEs in a simple linear setting.  Ultimately, they seek to make connections to recent work that has studied how over-parameterization (OP) plays a role in robustness.  They also provide a set of experiments to empirical study this setting.

**Questions:**

* The authors say that the $\ell_\infty$ and $\ell_2$ norms are "popular choices in practice."  What does "in practice" mean here?  I am not aware of any settings "in practice" that actually use $\ell_p$ robustness.  I think that in the broader context of the literature, $\ell_p$ robustness is seen as a stepping stone toward understanding more practically relevant security concerns, such as distribution shifts in autonomous vehicles tasks or medical imaging.

* In the sentence:

> "The compute needed to evaluate an image on the dense ViT models are 8.9 and 17.9 respectively."

What are the units on 8.9 and 17.9?

**Limitations:**

I did not see a discussion of this.  At the very least, the authors could comment on the limitations of their analysis.  Furthermore, they could comment on the potential for nefarious actors to use adversarial technology.  Although in principle I agree that this paper is theoretical in nature and that there aren't any concrete concerns from a societal point of view.

**Strengths And Weaknesses:**

### Strengths
---

**Similar linear setting.**  I liked the analysis that the authors performed for the case of linear MOEs.  Although the proofs seemed to follow standard techniques, I think that this part of the paper was a nice first step toward measuring the Lipschitzness of MOE models.

**New problem setting.** I have not seen many works that consider the problem setting of the robustness of MOEs.  If the robustness of MOEs is fundamentally different from that of other model classes, then this would represent a useful direction.


### Weaknesses
---

**Preliminaries** The authors repeatedly refer to the idea that MOEs have smaller computation cost.  I do not understand what this means.  Does it mean that the models are more computationally efficient at inference time?  Does it mean that they are less costly to train?  Without a description of how MOE models work and what their practical advantages are, I think that many readers will be left wondering the same thing.  For instance, they authors say that

> "Practical models also tend to this setup to match the FLOPs of dense models." (sic)

What does "FLOPS of dense models" mean?  Which dense models are the authors referring to?  What is even meant by a dense model -- e.g. convolutional neural networks, MLPs, etc?  I think this points to the need for a stronger preliminary section, where MOE and dense models are both introduced and defined.  Readers coming from the robustness literature may not be familiar with MOEs, so it's important to set those readers up so that they can understand the rest of the paper.

**Is this a problem?**  I'm having trouble discerning whether MOEs suffer from robustness issues.  The authors give some intuitive evidence for the lack of robustness of MOEs (e.g., that routing protocols can change given small perturbations), but there isn't any concrete  empirical evidence that MOEs are not robustness.  Therefore, I think it's not unreasonable to question why these results are meaningful.

Let me explain a bit more.  In the standard robustness literature, it has been repeatedly demonstrated that CNNs and MLPs are highly vulnerable to attacks.  For example, a PGD attack can degrade performance on CIFAR10 for ResNet18 from ~94% to essentially 0% (i.e., worse than random guessing).  This empirical result spawned all of the research concerning the robustness of these models.

In this paper, we are considering a new model -- MOEs -- which the authors describe as being more computationally efficient and using a fundamentally different design.  Therefore, I think that before studying the theoretical aspects of the robustness of MOEs, it would be worthwhile to demonstrate whether or not MOEs actually lack robustness to small perturbations.

**Related work.**  I think that the discussion of related work could be significantly improved.  At the very least, there should be a section called "Related Work" where the authors discuss past approaches for robustness and lines of work important to the development of MOE models.  Indeed, there seems to be previous work that studies the robustness of MOE models:

> Xu, Kaidi, et al. "Mixture of robust experts (more): A robust denoising method towards multiple perturbations." arXiv preprint arXiv:2104.10586 (2021).

The authors should cite works like these to distinguish their contribution from that of the literature.

**Claims in the introduction.**  Some of the claims made in the introduction seem to be inaccurate.  For instance, the claim that adversarial robustness only refers to studying *bounded* changes.  In fact, the field has evolved to study more general (non-norm-bounded) shifts in data; see e.g.,

> Wong, Eric, and J. Zico Kolter. "Learning perturbation sets for robust machine learning." arXiv preprint arXiv:2007.08450 (2020).

> Robey, Alexander, Hamed Hassani, and George J. Pappas. "Model-based robust deep learning: Generalizing to natural, out-of-distribution data." arXiv preprint arXiv:2005.10247 (2020).

> Hendrycks, Dan, et al. "The many faces of robustness: A critical analysis of out-of-distribution generalization." Proceedings of the IEEE/CVF International Conference on Computer Vision. 2021.

Furthermore, the authors claim that poor adversarial robustness is due to over-parameterization.  This also seems inaccurate, as poor robustness has also been attributed to adversarial subspaces

> Gilmer, Justin, et al. "Adversarial spheres." arXiv preprint arXiv:1801.02774 (2018).

and to the local linearity of DNNs with piece-wise linear activations

> Goodfellow, Ian J., Jonathon Shlens, and Christian Szegedy. "Explaining and harnessing adversarial examples." arXiv preprint arXiv:1412.6572 (2014).

among other causes.  These claims highlight the need for a more comprehensive literature review in my opinion.

**Connections to Bubeck and Selke [3].**  I don't understand the impact of the connections to [3].  Essentially, the argument seems to be that as MOEs have $E$ times more parameters, one can replace $P$ by $P\cdot E$ and then you get a constant factor in the bound.  The thing is, while we may consider taking the number of data points to infinity, we will never have very large number of experts in the same way from my understanding.  Sure, if you take $E\to\infty$, the bound gets small, but is this practically possible?

**Lack of a practical takeaway.**  One question I have about this work: What is are practical takeaways?  The goal of this paper seems to be to study the robustness of MOEs.  However, it's hard to draw any concrete conclusions based on my reading.  For example, does the theory inform the reader of any algorithm concerning how to improve MOE robustness?  Are there settings where MOEs are *significantly* more robust than ViT models?  The experiments seem rather non-conclusive to me in this regard, as the ViT models perform very similarly to the MOE models.  Indeed, in Figure 1, the curve are all almost overlapping.  The same is true in Figure 3.  So setting the theory aside, I'm not sure what the practical takeaways are from this paper.

**Proof of Lemma 1.**  I believe that there is an error in the proof of Lemma 1.  The proof begins by calculating the gradient of the mixture of experts:

$$ \nabla f(x) = \sum_{i=1}^E p_i(x) \cdot \nabla f_i(x) + \sum_{i=1}^E \nabla p_i(x) \cdot f_i(x)$$

We then take the norm of both sides, and apply the triangle inequality:

$$ || \nabla f(x) || \leq \sum_{i=1}^E p_i(x) \cdot ||\nabla f_i(x)|| + \left|\left| \sum_{i=1}^E \nabla p_i(x) \cdot f_i(x) \right|\right|. $$

Next we take the supremum over $x$ on both sides, yielding

$$ \text{Lip}(f) \leq \sup_{x} \sum_{i=1}^E p_i(x) \cdot ||\nabla f_i(x)|| + \sup_{x}\left|\left| \sum_{i=1}^E p_i(x) [s_i - \bar{s}(x)] \cdot f_i(x) \right|\right|. $$

We can further upper bound the RHS by moving the supremum inside the sum, meaning that

$$ \text{Lip}(f) \leq  \sum_{i=1}^E \sup_{x} p_i(x) \cdot ||\nabla f_i(x)|| + \sup_{x}\left|\left| \sum_{i=1}^E p_i(x) [s_i - \bar{s}(x)] \cdot f_i(x) \right|\right|. $$

So far, this aligns with the proof given in the paper (although I've written out the steps in more detail here).  This is the point at which I feel problems start to arise.  For any index $i\in[E]$, consider the expression

$$ \sup_{x} p_i(x) \cdot ||\nabla f_i(x)||  $$

The authors assert that this expression is less than or equal to (or simply equal to -- the proof is ambiguous here) $p_i(x) \text{Lip}(f_i)$, which I believe is incorrect (although if I'm wrong, I'd welcome a discussion with the authors).

In general, my intuition for this being wrong is that in general, for real-valued functions $q(x)$ and $g(x)$, it does **not** hold that $\sup_x(q(x) \cdot g(x)) = q(x) \cdot \sup_x(g(x))$ $\forall x$.  (In this particular case, we have $q(x) = p_i(x)$ and $g(x) = ||\nabla f_i(x)||$.)  To make this more concrete, let's consider a counterexample.  Let $f(x)$ be defined piecewise (here I've dropped the $i$ subscript):

$$ f(x) = (-1/2)x + 1/2 \quad\text{for}\quad x<-1 $$
$$f(x) = -x \quad\text{for}\quad -1\leq x\leq 1$$
$$f(x) = (-1/2)x - 1/2 \quad\text{for}\quad x>1$$

The Lipschitz constant of this function is clearly $\text{Lip}(f)=1$.  Now let

$$ p(x) = \epsilon \quad\forall x : |x| \leq 1$$
$$ p(x) = 1-\epsilon \quad\forall x : |x| > 1$$

where again, I've dropped the $i$ index for simplicity.  Then we have that

$$ p(x) ||\nabla f(x)|| = \epsilon \quad \text{for}\quad |x| \leq 1$$
$$ p(x) ||\nabla f(x) || = (1-\epsilon)/2 \quad\text{for}\quad |x| >1 $$

and therefore $\sup_x p(x) ||\nabla f(x) || = (1-\epsilon) / 2$ provided that $\epsilon < 1/3$.  On the other hand, we have that

$$ p(x) \text{Lip}(f) = \epsilon \quad\text{for}\quad |x|\leq 1 $$
$$ p(x) \text{Lip}(f) = 1-\epsilon \quad\text{for}\quad |x| > 1 $$

Therefore, we find that

$$ \epsilon = p(x) \text{Lip}(f) < \sup_x p(x) ||\nabla f(x)|| = (1-\epsilon)/2 $$

if $\epsilon < 1/3$ and $p(x) = \epsilon$ (since $p(x)$ is a function of $x$, which is a free variable, we can pick $p(x)$ to be whatever we like from the above definition).  Therefore, it does not hold that $\sup_x p(x) ||\nabla f(x)|| \leq p(x) \text{Lip}(f)$, breaking the proof of Lemma 1.

As several results and much of the discussion in the paper follow from this Lemma, I think it's important that this concern be addressed.

**Grammar.**  The grammar and structure of the paper could also be revised.  There are several typos as well, which should be addressed.

### Final thoughts
---

Overall, while I think that this paper goes in an interesting direction, I also believe that it would need to be fundamentally rewritten and expanded to be considered for acceptance.  The motivation for this problem is shaky, given that it is not clear whether MOE models lack robustness in the first place.  The literature review is sparse, and some of the proofs appear to be incorrect.  Furthermore, the experiments are not convincing, as the difference in robustness between dense and non-dense models doesn't seem to be large.  For these reasons, I cannot recommend accept at this time, especially given the possibility that the proof of Lemma 1 is incorrect.

---

> ### Author Response · Authors · 2022-08-02
> **Author response**
>
> We thank the reviewer for detailed comments.
>
> **Lemma 1:** We would like to first clarify that our proof of Lemma 1 doesn’t use the inequality reviewer mentioned and is in fact clear of the bug.
>
> Please note that the proof follows these steps for the final bound.
> 1. $\sum_{i=1}^k p_i(x) \| \nabla f_i(x) \| \leq \sum_{i=1}^k p_i(x) L_{f_i}$ .This follows from just upper bounding $\| \nabla f_i(x) \| $with $L_{f_i}$.
> 2. $\sum_{i=1}^k p_i(x) L_{f_i}  \leq \max_i { L_{f_i} }$. This just follows from Holder’s inequality and $\sum_{i=1}^k p_i(x) =1$.
> 3. We finally take $\sup_x$ on both LHS and RHS after this. Note that there is no $\sup_x$ before this step.
>
>
> **MoE preliminaries:** This is an oversight on our part and we will clarify the following MoE notation in the updated version of the paper.
>
> Dense model - The name dense comes from the fact that every parameter is used on every example. A dense model is equivalent to a MoE with number of experts (E) set to 1, i.e. a single f_i(x) function in equation 1 from the paper. For MoE models used in the experiments section of the paper, their dense counterpart is the standard Transformer model (ViT) with a single (E=1) MLP layer in each block.  The main advantage of MoE models is that they allow for more parameters/capacity (by setting E >1) but at roughly the same computation cost as a dense model with E=1 (when assuming a sparsity K=1 for the MoEs).
>
> **Lack of robustness of MoE:** MoEs differ from dense models primarily in the following way -  A dense model computes the same function for all its inputs. However a MoE model computes different functions for different inputs depending on the routing layer. This introduces robustness challenges to MoEs as perturbations to inputs can lead to different functions being applied to the input. We disagree with the statement that we do not provide “concrete empirical evidence”: Indeed, we demonstrate in Figure 2 that real world MoE models suffer from this issue and perturbations can lead to routing changes for more than 50% of the inputs! This alone suggests that MoEs could be much more brittle than dense models and makes it a worthwhile question to study when MoEs are more/less robust than dense models. However, our paper shows the surprising result that even though perturbations cause significant routing changes, MoEs trained in practice are robust in terms of final predictions/accuracy (Figure 1).
>
> **Related work:** Given the page limitation, we have focused on previous works that are most directly related to our problem of interest. We agree with the reviewer and have modified the introduction accordingly to account for the suggested related works.
>
> **Connections to Bubeck and Selke [3].** We note that our work is directly motivated by the lower bound on the Lipschitz constant presented in [3]. Our paper provides settings under which MoE models can achieve an upper bound that matches the 1/\sqrt{E} - scaling with the number of experts (E). Further, in practice, people have already experimented with MoE models with a number of experts of size 2048 (1/\sqrt{E} ~ 2.2e-2) (Lepikhin et al., 2020 [15]). Note that this observation is independent of the number of data points.
>
> **Practical takeaways.**
> Below are some high level practical takeaways from this work.
> 1) We show that MoEs can achieve better robustness accuracy (>5%) than dense models for a similar computational cost. We believe our work can inform future research to further explore the Pareto frontier of robustness vs. computational cost.
> 2) In MoEs, adversarial robustness has multiple facets, in terms of both routing assignments and final predictions (the latter being classically studied in dense models). Our work shows that MoEs have robust predictions despite fragile routing assignments. Practically, this indicates that the routing algorithms currently employed (e.g., noisy top-K) do not harm the final prediction
> robustness, in spite of their apparent non-smooth behaviour. We think this practical observation can inform future research to investigate whether more robust routing algorithms can further improve the final prediction robustness of MoEs.
>
> **Experiments:**  We would like to first clarify that before our work we are not aware of any study experimentally evaluating the robustness of MoE models and comparing them to dense models. In our experiments we evaluate the robustness of MoE models and show that on both JFT and Imagenet datasets, the standard V-MoE models achieve significantly better robust accuracy (>+5%) than the dense ViT model with the same computation cost (Figure 1).
>
> The dense model (ViT++) that achieves similar robust accuracy to the V-MoE model in Figure 1, is a much more expensive ViT model (ViT++) that has roughly 1.5x computation cost as the V-MoE model.

---

> > ### Comment · Reviewer_wLgw · 2022-08-09
> > **Ahhh Holder's Inequality, you win again**
> >
> > Hello authors,
> >
> > **Lemma 1.** Thanks for the clarification about the proof.  You never want to be _that_ reviewer who can't see Holder's inequality when it's staring them right in the face.  But today I am that reviewer.  The proof of Lemma 1 is correct.  This was the main reason for giving a low score, so I will be happy to adjust my score accordingly.
> >
> > **Prelims.**  I think updating the text regarding the MOEs prelims will make the paper stronger.  Thanks for your explanations.
> >
> > I think it still remains unclear (at least to me) what it means for these models to be more _computationally efficient_.  Perhaps this is simply my lack of understanding regarding MOEs.  But I feel that other readers may agree that this point could be made more clearly.
> >
> > **Figure 2.**  Indeed, this fact that MOEs are so vulnerable -- while perhaps underemphasized in the paper -- is clear from Figure 2.  Perhaps this should point be reiterated, because I certainly missed it while I was reading.  In general, I think it would help readibility to decouple the results showing the effectiveness of your method from those that demonstrate that MOEs are vulnerable.  I.e., you could have a section at the beginning of the experiments (or even at the beginning of the paper) which show just how vulnerable MOEs are.  Then later on you could show how your solution resolves the problem.  This is just one idea, but in general I see the authors' point, and agree that based on Figure 2, there is a significant vulnerability.
> >
> > **Bubeck and Sellke.**  I still don't really see the connection.  Perhaps it would help to experiment with large numbers of experts (such as the paper cited in your response above).  My confusion still centers around that in general we wouldn't think about taking $E\to\infty$ in practice.
> >
> > Overall, I think that the rest of the changes that the authors made have been positive.  I'll admit that I made a mistake when checking the proofs, and therefore my score was much lower than it otherwise would have been.  I will update accordingly.

---

### Official Review · Reviewer_XB3a · 2022-07-13

**Rating:** 5
**Confidence:** 3
**Soundness:** 2 fair
**Presentation:** 2 fair
**Contribution:** 2 fair

**Summary:**

The paper investigates the adversarial robustness of sparse Mixture of Experts models for image classification, in order to answer the question: do models with more parameters (but not necessarily more computational power) have better robustness? Bounds on the Lipschitz constant are derived to compare the sparse MoE models with their dense counterpart, to show that sparse MoE models are more robust. The claim is then verified with empirical experiments on large data sets.


**Questions:**

* In Section 3.2, the setup of the analysis is a regression problem instead of a classification problem. How do the results presented are impacted by the actual setup of ViT-MoE, where we have a non-linear model with multiple MoE layers in a classification problem? The layers between the MoE layers could amplify the impact of the perturbations and reduce the "smoothness" gained by using MoE.
* In Definition 1, the statement is trivial. There's no restriction in $\epsilon_1$, so it could be any arbitrarily large value. The definition is just saying that there is a number larger than another number.
* In Section 4, the term "token" is used, but it is never described in the paper.
* From Figure 1, ViT++ and ViT-MoE are indistinguishable. Is the improvement in robustness coming from the MoE or just from the increased model size?
* Related to Figure 2: the figure has the rate of changes per layer. It would also be interesting to see if the attack always modifies the routing in all layers or if there is some pattern (e.g. for some inputs the change only happens in later layers; or, can the change in one layer have no impact on the next?)
* In Figure 3, again, how significant is the difference? How does MoE helps with robustness?
* Adaptive attacks: in Section 4.4, the auxiliary losses are included in the attack. What are those losses and how are they optimized? Does the attacker specifically tries to change the output of the routing? Can an attacker force all images to be routed to a single expert?
* Please expand the discussion of related works, for example: [1, 2].
* Improve the mathematical description in Section 2:
    - Add the dependency on the label $y$ in the formulation of the adversarial examples in eqs 3, 4, 5.
    - line 115: the clarification after "i.e." is actually confusing (is the 0/1 loss used for evaluation only or for generation of the examples in the preceding equation?)
    - Eq 5: in the current text the variable $t$ is defined as a real number. Replace with $\forall t \in \{0, \dots, \tau -1\}$

### References
[1]: Wu, B., Chen, J., Cai, D., He, X., & Gu, Q. (2021). Do Wider Neural Networks Really Help Adversarial Robustness? https://arxiv.org/abs/2010.01279
[2]: Xu, K., Wang, C., Cheng, H., Kailkhura, B., Lin, X., & Goldhahn, R. (2021). Mixture of robust experts (more): A robust denoising method towards multiple perturbations. https://arxiv.org/abs/2104.10586

**Limitations:**

The main limitation of the work is how applicable their theoretical analysis is to the actual large models of interest. It is not clear how the limited setup of their analysis actually applies to more complex models as used in the empirical experiments.

**Strengths And Weaknesses:**

## Strengths
+ The topic of the paper is relevant, both from the perspective of robustness analysis (what makes a model robust?) and the MoE perspective (with trend towards large models, MoE is a viable alternative, so what other benefits we get besides computational cost?).
+ The paper has a very clear goal: are sparse MoE models more robust than their dense counterpart?
+ The paper is well organize and, generally, it was easy to read.
+ The empirical experimentation is performed on very large data sets and non-trivial models.
+ Originality: while there are papers investigating robustness of large models, they do not analyze MoE models.

## Weaknesses
- The mathematical expressions/descriptions are sometimes imprecise/incomplete (e.g. from the eqs in Sec 2.2, it would seem that the true label of the image is not used to generate the adversarial example). See more in the questions.
- The theoretical analysis (Sec 3) focus on linear (single layer) experts for a regression task and fixed routing. There's little discussion on how these results transfer to the much more complex setting of ViT-MoE (non-linear, multiple layers of MoE, non-fixed routing) for classification.
- In the empirical evaluation, it is difficult to distinguish improvement (in particular for JFT-300M) and the significance of the results
- Part of the analysis is conducted in a private data set (JFT-300M), which makes impossible to reproduce those results.
- Limited discussion of related works

## Summary
Overall, the paper is clear, easy to follow and pursuits an original analysis of an interesting and relevant problem. However, it needs to be polished more as there are several issues that put in question the quality and significance of the results.

---

> ### Author Response · Authors · 2022-08-02
> **Author response**
>
> **Theoretical analysis.** Our goal in the paper is to compare the robustness of MoE with dense models (# experts E=1), and we present this comparison by analyzing the linear models. We agree that having such a comparison for deep neural networks is an interesting direction. However please note that there has been limited theoretical analysis of Lipschitz constant of deep networks and considering linear models is common in the literature [1-5]. Further we analyze the mixture of expert models which introduces another complication for analysis. Given there has been no work on analyzing the MoE networks, we believe our work takes a good first step towards presenting factors that affect the robustness of MoEs.
>
> Regarding the classification setting, even for linear models, studying the classification regime is challenging and it is still an active research topic (e.g. [6]), with limited results for neural networks. As a result, not to add an extra layer of complexity, we believe that addressing the regression setting is a fair starting assumption.
>
> 1. [https://arxiv.org/abs/1810.11914](https://arxiv.org/abs/1810.11914)
> 2. [https://arxiv.org/abs/2007.15220](https://arxiv.org/abs/2007.15220)
> 3. [https://arxiv.org/abs/1804.11285](https://arxiv.org/abs/1804.11285)
> 4. [https://arxiv.org/abs/2104.09437](https://arxiv.org/abs/2104.09437)
> 5. [https://proceedings.mlr.press/v125/javanmard20a.html](https://proceedings.mlr.press/v125/javanmard20a.html)
> 6. [https://arxiv.org/abs/0910.4627](https://arxiv.org/abs/0910.4627)
>
> **Experiments:**  We would like to first clarify that before our work we are not aware of any study experimentally evaluating the robustness of MoE models and comparing them to dense models. In our experiments we evaluate the robustness of MoE models and show that 1) on both JFT and Imagenet datasets, the standard V-MoE models achieve significantly better robust accuracy (>+5%) than the standard dense ViT models. 2) We further show that this trend remains even after adversarial training. 3) Finally we show that one requires a much more expensive ViT model (ViT++) to achieve similar robust accuracy than the standard V-MoE model.
>
> **Token.** Token refers to an element in the sequence input to the models in Section 4. Please note that unlike Resnets, input to Transformer models is a sequence of tokens, where each token is a vector. ViT models convert each image patch into a token vector using a linear projection layer (see Fig 1 of https://arxiv.org/abs/2010.11929). Hence an image is first converted to a sequence of image patches, which are in turn converted into a sequence of tokens.
>
> **Figure 1.** First we show in the figure that ViT-MoE has significantly better robust accuracy than the dense ViT models. To check if the improved robustness is a function of the MoE architecture or the model size, we train a much more expensive dense ViT++ model, and show that one can achieve similarly robust accuracy as the MoE but with much more computation (+44.4% more FLOPs).
>
> **Figure 2.** That is an interesting suggestion. We will include such an analysis in the final version.
>
> **Figure 3.** In the leftmost figure we see a significant gap (>5%) in robust accuracy between the dense ViT and the MoE models for \epsilon of O(1e-3).  As we increase \epsilon all models collapse to zero robust accuracy, which is expected for models without adversarial training.
>
> **Auxiliary losses:** Each MoE layer has auxiliary losses to encourage load balancing between experts, i.e., to encourage an equal number of examples are routed to all experts. Our attack maximizes this loss in addition to the final classification loss. Maximizing this auxiliary loss in the adversarial attack encourages less balanced routing. It is possible for the routing to completely collapse and send all examples to a single expert, and is a solution of the maximization.
>
> **Section 2.** We will update loss to include the true label ($y$) in all equations.

---

> > ### Comment · Reviewer_XB3a · 2022-08-09
> > **Increased score, but still think paper fails to make a strong and convincing argument.**
> >
> > Thank you for the reply. Your comments have improved my perception of the paper and I have increased my score.
> >
> > However, I still believe the paper needs more polish and even after reading the authors' reply, I still find that the paper fails to present a strong and convincing argument in towards its goals.
> >
> > The main question of the paper is: Are large MoE more robust? The theoretical analysis claims that the answer is yes, but the presentation of the results struggles to make a good argument. The paper is motivated through the lens of deep neural networks. But the main theoretical results are derived in a very different setting and the presentation doesn't make it clear the presented results are relevant, in particular from the motivation of large deep neural networks where the Lipschitz constant could be arbitrarily impacted from one layer to the next.
> >
> > I would also suggest that a focused discussion of related work could help situate the contributions of this paper, both in terms of the focus on MoE approach as well as in the context of robustness analysis via Lipschitz constants.
> >
> > And when it comes to the empirical results, the message is still muddled there. The main robustness gains are for very small perturbations. When it comes to the usual range of $\epsilon = 8/255   (\sim0.031)$ or stronger (which is the more common setting in adversarial papers), there is no clear gain in robustness. The lack of error bars also makes it harder to assess the true significance of the results, even where the gains are more pronounced. Again, I think the message is not clear. Most of Figure 3 is not really backing up the claims. And in particular, the gain of MoE from the computational perspective (as was discussed in the reply above) is not made clear from the Figure.
> >
> > I appreciate the author's response to my concerns and other reviewers' questions and the contribution and novelty of the work is better explained after the comments, which is why I increased my score. But overall, I still think the paper struggles to make a convincing case, which is why I still think it is borderline.

---

### Official Review · Reviewer_uHiB · 2022-07-16

**Rating:** 5
**Confidence:** 4
**Soundness:** 3 good
**Presentation:** 3 good
**Contribution:** 3 good

**Summary:**

The paper focuses on the adversarial robustness of mixture of expert (MoE) models and proposes theoretical analysis focusing on the Lipschitz constant of an MoE model under some relaxed assumptions such as a smooth MoE and a linear model.
The goal is to answer the interesting question of whether MoE models, with significantly higher number of parameters and roughly constant additional computation cost, have better adversarial robustness than their dense (single-model) counterparts.
The analysis shows that under certain conditions on the routing function and the structure of the data, MoE models can have a significantly lower Lipschitz constant. Under some ideal conditions, the lower bound on the Lipschitz constant established by Bubeck and Sellke [3] for single over-parameterized models can be reduced by a factor of $\sqrt{E}$ for a mixture of $E$ experts.
They experimentally evaluate the adversarial robustness of the recent Vision Transformers and Vision MoE models on a proprietary dataset (for pre-training) and the ImageNet dataset (for fine-tuning), and report interesting observations about the robustness of MoE models, the choice of experts, and redundancy among the experts.


**Questions:**

Please find a list of my questions and comments below. Major ones are listed first.

**1:** For the case $\mathbf{X}_1 \\,\bot\\, \mathbf{X}_2$, it is not clear to me how to go from
$\\| (\mathbf{X}^T \mathbf{X})^\dagger \mathbf{X}^T \mathbf{y}  \\| \\,$ to $\\, \\| (\mathbf{X}_1^T \mathbf{X}_1)^\dagger \mathbf{X}_1^T \mathbf{y}_1 \\,+\\, (\mathbf{X}_2^T \mathbf{X}_2)^\dagger \mathbf{X}_2^T \mathbf{y}_2  \\| $. Maybe I missed some trick or detail.

**2:** Could the authors clarify the statement on lines 196 - 199 about the construction of the projection matrices and also specify their dimension?

**3:** Lines 242 - 243: it is mentioned that JFT-300M is a multi-label dataset and uses the sigmoid cross-entropy loss. By multi-label, does it mean each sample has 1 or more labels, or does it simply mean that there are multiple classes, with each sample assigned to a single class? Also, does sigmoid cross-entropy loss mean the same as binary cross-entropy loss (which would only be suitable for binary classification)? Please clarify how many classes are present in JFT-300M and provide some additional details on the dataset (that is reasonable to share).

**4:** Please provide a formal  definition of the following metrics (in the appendix if space is limited): Precision at 1, False discovery rate, and Intersection-over-union.

**5:** In the results in Figure 1, the metrics reported are referred to as “classification error rate” and “false discovery rate”. It would be more accurate to call them “robust error rate” and “robust FDR” respectively because these are considering the worst-case adversarial input $x’$ for each clean input $(x, y)$. In other words, the metrics would be defined with a `max` over a $\ell_\infty$ norm ball. It is misleading to refer to them as a standard error rate or FDR both in the figures and the text. Same comment for Figure 3.

**6:** In figure 3, it is mentioned that dashed lines are used to indicate that the auxiliary losses are also being attacked. However, I do not see any curves with dashed lines.

**7:** Referring to figure 3 (rightmost figure) and lines 320 - 322, it is mentioned that for finetuning on ImageNet, all models with more than 2 experts achieve roughly the same accuracy (this should be robust accuracy). Can the authors provide some insight on why there is no notable improvement in the robustness (robust accuracy) as the number of experts increases?

**8:** In the interest of completeness and reproducibility, please define all the attack objectives including the ones against the auxiliary losses in an appendix. Please define the auxiliary losses used to balance the load among the experts.

**9:** In section 4.4, it is mentioned that the result for the classification error on ImageNet has a similar trend. This result can be included in an appendix.

**10:** In figures 1 and 3, it would be easier to see the contrast in metrics if the x-axis (perturbation size) is restricted to smaller value (say 0.01). The y-axis could be log-scaled if that helps.

**11:** There are a few places where it stated (for example) that the classification error is x and y respectively, but it is not clear which two models the values refer to. For example, on lines 260 - 262, the classification error of which two models are 19.3% and 17.8%? Same comment for the false discovery rate.

**12:** In lines 85, 86 it is mentioned that only one expert is selected for each input, i.e. $K = 1$. Where is this assumption used?

**13:** In equations 3, 4, and 5, the loss function in the attack objectives should have a class label or target $y$ (currently missing). Also, a distinction should be made whether it is a untargeted or targeted attack.

**14:** In Lemma 1 and other places, please clarify what type of norm is being used. Or state upfront that the norms are by default (e.g.) $\ell_2$.

**15:** On line 186, I think it should be $\mathbf{X}^T = [\mathbf{X}_1^T ~\mathbf{X}_2^T]$ in order to have the right dimensions.

**16:** Nit: On line 142, it should be $\\{ \mathbf{s}_i \in \mathbb{R}^D \\}_\{i \in [E]\}$. It seems like the set of $\mathbf{s}_i$ lives in $\mathbb{R}^D$.

**17:** On line 168, I think it should be equation 7, not 6.

**18:** Nit: Hyphen missing in definitions 1 and 2. Should be “In-subspace distance” and “Cross-subspace distance”.

**19:** Lines 237 - 238: Please clarify that the 8.9 GFlops corresponds to the ViT-B/32, while the 17.9 GFlops corresponds to the ViT-B++/32 model.

**20:** A table with the compute in GFlops and the (order of) number of parameters for the different architectures, as well as for different number of experts would be useful. Please have a forward reference to Table 1 in the Appendix.

**21:** Please clarify what is meant by a Token.



**Limitations:**

The authors discuss some limitations of their work in section 5. One of them is the focus on linear models for simplicity of analysis. Additionally, the paper focuses on the analysis of smooth mixture of experts, whereas in practice sparse MoEs are not smooth or continuous. This should be discussed under limitations.


**Strengths And Weaknesses:**

### Strengths
**S1:** The paper focuses on the theoretical and empirical analysis of the adversarial robustness of mixture of expert (MoE) models. This is an important problem since the robustness of neural networks to bounded adversarial perturbations is a desirable property.

**S2:** The work seems to be relatively novel to my knowledge since existing works have not focussed much on the analysis of robustness of over-parameterized MoEs through the lens of Lipschitz constant. It is inspired by the recent work of Bubeck and Sellke [3], who establish a result on the lower bound of the Lipschitz constant of over-parameterized neural networks in terms of the number of parameters, samples, and the dimension.

**S3:** The presentation of technical details is fairly clear to follow, except in a few places where some clarifications would help (please see my questions).

### Weaknesses

**W1:** Given that practical MoEs follow the sparse combination of experts as specified by equation (1), they are not continuous functions. Hence, small changes in their input can lead to large changes in their output. The analysis presented in the paper assumes a smooth weighted mixture of experts. It is not clear how this analysis can be extended to the sparse MoEs, or how one can draw insights from the analysis for the case of sparse MoEs.

**W2:** Some of the details about the experiments are not clear or missing (e.g., attack objectives, definition of metrics). This would be important to make the work more reproducible (please see my questions).

**W3:** (Minor) The analysis could be supported with some intuitions and maybe a figure (a toy example) for illustration. That would convey the key ideas in a better way.

---

> ### Author Response · Authors · 2022-08-02
> **Author response**
>
> **W1**- Our analysis of smooth MoEs presents conditions on 1) how data is routed to experts and 2) stability of the router that results in robust MoE models. We note that the first component is the same even for sparse MoEs. However router stability can be different for sparse MoEs. We believe that noisy routing—a popular technique that adds noise to routing probabilities—will make sparse MoEs more similar to smooth MoEs (see e.g. [https://arxiv.org/abs/2002.08676](https://arxiv.org/abs/2002.08676) for a broader analysis of non-differentiable operators by smoothing with noise injection), and is an interesting direction for further research. Theoretical analysis of MoEs is lacking in general and ours is the first step in that direction. In addition to that, in order to close the gap between our theoretical analysis and practical models, we also provide experimental evidence that, indeed, sparse V-MoE models present the same robustness properties as their dense counterparts, at a much smaller computational cost.
>
> **W2**- We use standard adversarial attacks and metrics in our experiments, widely used in many other papers in the field. In any case, we will add additional information to clarify our setting in the appendix, and we will open source all the software used to conduct the experiments described in the paper.
>
> **Questions**
> 1. We are mainly using 1) $X^T X = X_1^T X_1 + X_2^T X_2$, 2)  $(X_1^T X_1 + X_2^T X_2)^\dagger =  (X_1^T X_1)^\dagger + (X_2^T X_2)^\dagger$, and 3) $(X_1^T X_1)^\dagger X_2^T = 0,$ where 2 and 3 follow from from orthogonality of $X_1$ and $X_2$. We will detail these steps in the final version.
>
> 2. Projection matrices $U_i \in \mathbb{R}^{D \times D_i}$, where $D_i$ is the dimension of the subspace spanned by $X_i$. Columns of $U_i$ are the singular vectors of $X$ with lowest subspace distance to $X_i$ in comparison to $X_j, j \neq i.$
>
>
> 3. Each sample in JFT-300M can have more than one correct class label. Hence, for each label, we use a binary cross entropy loss (with a sigmoid link function) as in earlier works using this dataset [6, 19]. The dataset has more than 18000 classes. Each image has about 5 labels on average.
>
> 6. In Figure 3, dashed lines are overlapping with the solid lines making them hard to differentiate. This establishes that attacking auxiliary losses has little effect on the adversarial robustness in terms of accuracy.
>
> 7. Unlike JFT, when finetuning on Imagenet, we notice that even standard accuracy does not improve with more experts. We hypothesize this is because ImageNet being a relatively easier task than JFT300M and we thus observe a saturation earlier as we increase the model capacity with more experts. We observe that robust accuracy also follows this same trend as the standard accuracy.
>
> 8. We use the same auxiliary losses from the V-MoE paper (Section A.2 in [19]) without any changes to the model. We will add these details in the appendix for completeness.
>
> 11. In lines 260-262, we are referring to ViT-B/32 and V-MoE-B/32 models achieving 19.3% and 17.8% classification error rate on ImageNet, respectively.
>
> 12. The assumption of $K=1$ is used in the definition of sparse MoE, in Eq. (1). However, in the experiments section we use $K=2$ because that’s the value used in the original V-MoE paper. We will update this paragraph in the text to avoid any confusion.
>
> Thanks for the suggestions 4, 5, 9, 10, 13-20. We will update the final version to address them.

---

> > ### Comment · Reviewer_uHiB · 2022-08-09
> > **Follow-up to author response**
> >
> > Thank you for the response. I could not find a revised submission that addresses the issues pointed out by the reviewers. Although the response has addressed some of my concerns, the paper is currently not very polished. On the positive side, the work is novel and has technical depth. However, there are still missing details on the experiments (auxiliary losses, attack objectives etc) as pointed out in W2 of my review. The content in Appendix D and E are never addressed in the main paper. I also share some of the concerns raised by Reviewer `XB3a`. For these reasons, I am lowering my score to 5.

---

> > > ### Author Response · Authors · 2022-08-09
> > > **Updated manuscript and supplemental material**
> > >
> > > Dear Reviewer uHiB,
> > >
> > > We intended to update the manuscript after the additional experiments that Reviewer P3QL suggested were completed. In the meantime, we have updated the manuscript with several of the points that we promised to address (in particular, the definition of the auxiliary losses and metrics used for the evaluation).
> > >
> > > We appreciate your comments about the novelty and technical depth of the work (shared with the rest of reviewers), and hope that our changes (and those yet to come after the additional experiments are completed) help to reconsider your score.

---

> > > > ### Comment · Reviewer_uHiB · 2022-08-10
> > > > **Revised submission**
> > > >
> > > > Dear authors:
> > > >
> > > > I will take into consideration your revised submission. Thanks for your efforts.

---

### Author Response · Authors · 2022-08-09
**Updated manuscript**

Dear reviewers,

We have updated the manuscript and the supplemental material with several of the changes that we promised during the rebuttal phase. Some other changes are still pending of experiments to complete (e.g. providing additional results with adversarial attacks other than PGD), and we will update the manuscript as soon as we get those results.

We hope that these changes make the paper clearer and successfully address the points raised by the reviewers.

---

### Meta-Review · Area_Chair_N9QU · 2022-08-23

**Recommendation:** Accept
**Confidence:** Less certain

**Metareview:**

The paper studies the adversarial robustness of Mixture of Expert models. The authors have addressed the concerns from the reviewers and the reviewers adjusted their scores. However, the reviewer still has some concern that the paper fails to present a strong and convincing argument in towards its goals.

Please revise the paper as suggested by the reviewers in the detailed reviews. Please also complete the pending experiments and add to the next version.


**Award:**

No

---

### Decision · Program_Chairs · 2022-09-14

Accept